# Endogenous fluctuations of OCT4 and SOX2 bias pluripotent cell fate decisions

Daniel Strebinger[†] ID, Cédric Deluz[†], Elias T Friman[†] ID, Subashika Govindan, Andrea B Alber & David M Suter[*] ID

## Abstract

SOX2 and OCT4 are pioneer transcription factors playing a key role in embryonic stem (ES) cell self-renewal and differentiation. How temporal fluctuations in their expression levels bias lineage commitment is unknown. Here, we generated knock-in reporter fusion ES cell lines allowing to monitor endogenous SOX2 and OCT4 protein fluctuations in living cells and to determine their impact on mesendodermal and neuroectodermal commitment. We found that small differences in SOX2 and OCT4 levels impact cell fate commitment in G1 but not in S phase. Elevated SOX2 levels modestly increased neuroectodermal commitment and decreased mesendodermal commitment upon directed differentiation. In contrast, elevated OCT4 levels strongly biased ES cells towards both neuroectodermal and mesendodermal fates in undirected differentiation. Using ATAC-seq on ES cells gated for different endogenous SOX2 and OCT4 levels, we found that high OCT4 levels increased chromatin accessibility at differentiation-associated enhancers. This suggests that small endogenous fluctuations of pioneer transcription factors can bias cell fate decisions by concentration-dependent priming of differentiation-associated enhancers.

**Keywords** differentiation; embryonic stem cells; endogenous protein fluctuations; OCT4; SOX2

**Subject Categories** Development & Differentiation; Stem Cells & Regenerative Medicine

**Mol Syst Biol. (2019) 15: e9002**

## Introduction

Embryonic stem (ES) cells can be maintained in a self-renewing state *in vitro* or be driven to commit to specific fates when exposed to differentiation signals. However, ES cells often exhibit asynchrony and divergences in fate commitment when subjected to the same differentiation cues. This obscures the interpretation of how instructive signals impact cell fate decisions, and limits the

generation of pure ES cell-derived cell populations for future regenerative medicine applications (Cohen & Melton, 2011). Heterogeneity in cell fate commitment points at the coexistence of different cellular states, but these remain largely uncharacterized at the molecular level. Intercellular variability in expression levels of cell fate regulators constitutes a potential source of variable cellular states. One well-studied example is the heterogeneity of NANOG expression in serum + LIF culture conditions, which reflects reversible transitions of ES cells between the naïve and primed states (Filipczyk *et al*, 2015). However, protein expression variability of cell fate regulators in ES cells maintained in a more homogeneous, naïve state is poorly explored.

The transcription factors SOX2 and OCT4 (also known as POU5F1) are expressed in ES cells and are strictly required to maintain their pluripotent state (Niwa *et al*, 2000; Zhao *et al*, 2004; Chew *et al*, 2005; Okumura-Nakanishi *et al*, 2005; Masui *et al*, 2007; van den Berg *et al*, 2010; Tapia *et al*, 2015). Recent studies reported that quantitative properties of SOX2 and OCT4 such as expression levels or DNA binding properties are predictive of cell fate decisions at the four-cell stage (Goolam *et al*, 2016; White *et al*, 2016). Differences in SOX2 concentrations were also shown to change its enhancer binding profile (Mistri *et al*, 2018); however, whether this translates into differences in cell fate commitment is unknown. SOX2 and OCT4 were also reported to play antagonistic roles in the differentiation of ES cells towards the neuroectodermal (NE) and mesendodermal (ME) fates (Zhao *et al*, 2004; Thomson *et al*, 2011). However, these conclusions were largely based on long-term overexpression/knockdown or indirect correlations from fixed cells. A recent study in human ES cells has shown that OCT4 levels are predictive for the cell fate choice between pluripotent self-renewal and extra-embryonic mesoderm commitment (Wolff *et al*, 2018). Nevertheless, how endogenous expression levels of SOX2 and OCT4 fluctuate over time in naïve ES cells and whether these fluctuations bias germ layer cell fate commitment remains unknown.

Here, we investigated how endogenous variability in SOX2 and OCT4 protein levels impact the ability of ES cells to differentiate towards the NE or ME fates. To do so, we generated knock-in mouse ES cell lines allowing to monitor endogenous fluctuations of SOX2 and OCT4 proteins in live cells. We found that both proteins

---

Sponsored Stem Cells Research Chair (UPSUTER), The Institute of Bioengineering (IBI), School of Life Sciences, Swiss Federal Institute of Technology, Lausanne, Switzerland
*Corresponding author. Tel: +41 21 693 96 31; E-mail: david.suter@epfl.ch
†These authors contributed equally to this work

fluctuate over a 2- to 3-fold range with timescales of approximately one cell cycle in naïve mouse ES cells. Endogenous expression levels of OCT4 and to a lesser extent of SOX2 at the onset of differentiation impact the ability of pluripotent cells to differentiate towards NE and ME upon undirected differentiation. Using ATAC-seq on cells with different endogenous OCT4 levels, we show that OCT4 fluctuations are associated with changes of chromatin accessibility of enhancers involved in cell differentiation.

# Results

### Generation of a sox2-SNAP/oct4-halo knock-in ES cell line

We first aimed at generating a cell line allowing monitoring of SOX2 and OCT4 protein levels in single living cells. To do so, we used CRISPR-Cas9 genome editing (Cong *et al*, 2013; Yang *et al*, 2013) to knock-in fluorescent tags in fusion to the C-terminus of endogenous SOX2 and OCT4 proteins. We generated a cell line in which both alleles of SOX2 are fused to a C-terminal SNAP-tag, and one allele of OCT4 is fused to a C-terminal HALO tag (Figs 1A and EV1A–C). Since SNAP and HALO tags allow orthogonal labelling with fluorescent dyes, endogenous SOX2 and OCT4 levels can be measured independently in individual living cells (Fig 1B). These knock-ins were generated in a previously established reporter cell line for ME and NE commitment (SBR cell line (Deluz *et al*, 2016)), thus allowing to monitor SOX2 and OCT4 protein levels in live cells and to track differentiation outcomes. The resulting cell line was named SBROS for Sox1/Brachyury/ Reporter/OCT4-HALO/SOX2-SNAP. Because of the heterozygosity of the OCT4-HALO knock-in, we assessed whether OCT4-HALO levels are a good proxy for total OCT4 levels at the single-cell level. To do so, we combined immunofluorescence against total OCT4 with direct labelling of OCT4-HALO using the HaloTag TMR dye. We found that total OCT4 and OCT4-HALO levels were well correlated in single cells, suggesting that OCT4-HALO levels allow estimating total OCT4 levels (Figs 1C and EV1D). We then verified whether this cell line expresses normal levels of pluripotency markers by quantitative immunofluorescence analysis. In contrast to Western blotting, immunofluorescence is not biased by differences in membrane transfer efficiency of proteins with different molecular weights (such as OCT4 and OCT4-HALO) and allows obtaining distributions of protein levels in the cell population. We found that the SBROS cell line expressed on average 89% of wild-type mean OCT4 levels, 111% of wild-type mean SOX2 levels and 129% of wild-type mean NANOG levels. The distributions and median expression levels of these proteins were similar to those of wild-type E14 ES cells (Fig 1D). We also found the half-lives of both OCT4-HALO ($7.8 \pm 1.3$ h) and SOX2-SNAP ($8.1 \pm 1.0$ h) to be close to published half-life values for these proteins (Fang *et al*, 2014; Pan *et al*, 2016; Liu *et al*, 2017a) (Fig 1E and Table EV1), and the average cell cycle length of SBROS and wild-type E14 ES cells were similar (Fig 1F and Table EV2). mRNA levels of pluripotency markers of SBROS cells were mostly unaltered (Fig EV1E and Table EV3). We also verified cloning efficiency and alkaline phosphatase activity after 1 week of clonal growth, and found it to be comparable between SBROS and its parental SBR cell line (Fig EV1F).

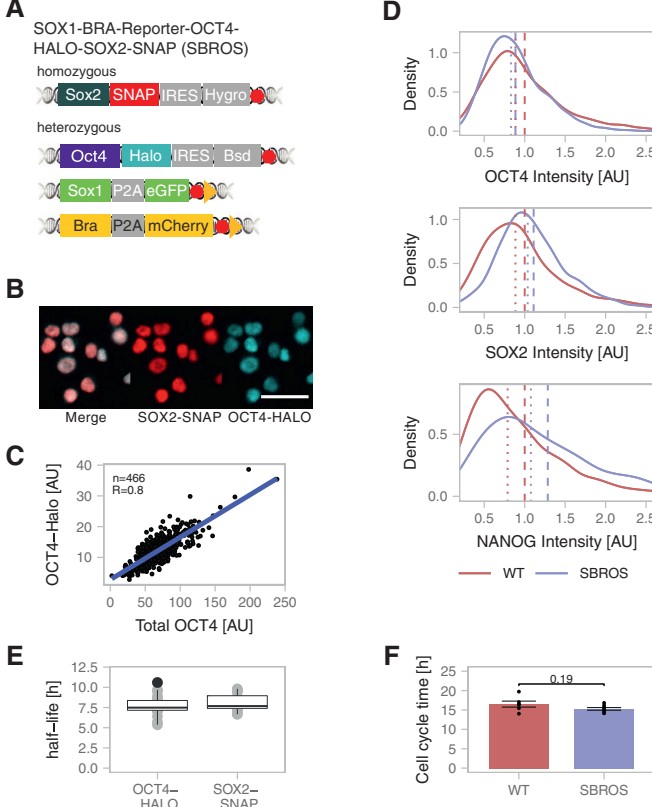

**Figure 1. Knock-in cell lines.**

A Scheme of the knock-in alleles of the SOX1-BRA-Reporter-OCT4-HALO-SOX2-SNAP (SBROS) cell line. Red hexagons: STOP codons. Yellow triangles: loxP sites.

B Example images showing the localization of SOX2-SNAP and OCT4-HALO by fluorescence microscopy of SBROS cells. Scale bar: 50 μm.

C Correlation between OCT4-HALO and total OCT4 levels determined by immunofluorescence and HALO labelling. *R* is Pearson's correlation coefficient.

D Distributions of OCT4 ($n = 2682$), SOX2 ($n = 1236$) and NANOG ($n = 2416$) levels in WT E14 and SBROS cell lines as determined by quantitative immunofluorescence. Dashed lines: mean protein levels; Dotted lines: median protein levels.

E Protein half-lives of OCT4-HALO and SOX2-SNAP in SBROS cells. Whiskers: minimum and maximum values; box: lower and upper quartiles; solid line: median; solid points: outliers; 1 biological replicate with $n = 20$ cells for each sample.

F Cell cycle duration of E14 WT ($n = 6$) and SBROS ($n = 8$) cells. Error bars: SE. *P*-value: two-sided *t*-test with unequal variance.

While the normal pluripotent phenotype of SBROS cells implies that the homozygous *Sox2-SNAP* alleles are functional, this is not necessarily the case for the heterozygous *Oct4-Halo* allele. To verify the functionality of the OCT4-HALO fusion protein, we drove its expression from the constitutive EF1-α promoter in Zhbtc4 cells, which allow doxycycline (dox)-inducible OCT4 knockout. After 24 h of dox induction, endogenous OCT4 expression is lost (Niwa *et al*, 2000), and thus, cells rely only on the OCT4-HALO protein to maintain their pluripotent state. In these conditions, we found that the OCT4-HALO protein was able to fully rescue pluripotency, thus confirming its functionality (Fig EV1G and Table EV4). We also

found the interaction of the OCT4-HALO protein with SOX2 to be preserved by co-immunoprecipitation experiments in the SBROS cell line (Fig EV1H), and the ChIP-seq profile of OCT4-HALO to be enriched at WT OCT4 peaks (Fig EV1I).

## SOX2 fluctuations regulate OCT4 levels

It has been reported that SOX2 and OCT4 protein levels are positively correlated in single cells (Filipczyk *et al*, 2015); however, the mechanism underlying this correlation is not understood. While ChIP-seq studies have shown that SOX2 and OCT4 bind to the regulatory sequences of the *Sox2* and *Oct4* genes (Chew *et al*, 2005; Loh *et al*, 2006), the functional impact of their binding on *Sox2* and *Oct4* expression is unclear. To determine how SOX2 and OCT4 impact their own and each other's expression levels, we established doxycycline (dox)-inducible ES cell lines allowing precisely timed expression of fusions of SOX2 or OCT4 to the YPet fluorescent protein. The functionality of these fusion proteins was validated by their ability to rescue pluripotency of inducible SOX2 or OCT4 knockout cells (2TS22C (Masui *et al*, 2007) or Zhbtc4 cells, respectively) (Figs EV1G and 1J, Tables EV4 and EV5), and by ChIP-seq analysis that showed genome-wide binding profiles similar to WT SOX2 and OCT4, respectively (Fig EV1I).

We then used these constructs to investigate the impact of YPet-OCT4 and YPet-SOX2 expression on endogenous SOX2 and OCT4 protein levels. To do so, YPet-OCT4 and YPet-SOX2 inducible ES cell lines were treated with dox for 0, 4 or 7 h, followed by immunofluorescence staining with anti-OCT4 and anti-SOX2 antibodies. While upon YPet-OCT4 overexpression, SOX2 levels remained stable (Fig EV2A–C), SOX2 overexpression modestly increased OCT4 levels already 4 h after induction of SOX2 overexpression (Fig EV2D–F). Interestingly, SOX2 levels slightly decreased as a function of overexpressed YPet-SOX2 over time (Fig EV2G), raising the possibility that SOX2 negatively regulates its own expression. Note that while some studies have suggested that SOX2 positively upregulates *Sox2* expression, these were either based on analysis of SOX2 binding to *Sox2* regulatory regions, which does not allow to infer the functionality of these interactions (Boyer *et al*, 2005; Chen *et al*, 2008), or using prolonged SOX2 depletion using siRNA (Chew *et al*, 2005) that compromises the pluripotent state (Masui *et al*, 2007). Furthermore, our findings are in line with previous studies investigating the functional impact of SOX2 on *Sox2* mRNA expression (Kopp *et al*, 2008; Ormsbee Golden *et al*, 2013).

To confirm negative SOX2 autoregulation, we then monitored endogenous SOX2 levels in live cells upon exogenous SOX2 expression in another knock-in cell line in which both endogenous *Sox2* alleles are fused to *nanoluc*, allowing real-time monitoring of endogenous SOX2 levels by luminescence microscopy. Additionally, a P2A-Firefly Luciferase (FLUC) cassette was also knocked-in in fusion to one allele of *Sox1* to monitor NE commitment (Figs EV2H and EV1A–C). We called this cell line Sox2-Nanoluc-Sox1-Fluc (SNSF). Quantitative immunofluorescence analysis confirmed that the NANOLUC tag did not strongly alter expression-level distributions of pluripotency factors, as cells retained 86% of wild-type mean OCT4 levels, 79% of wild-type mean SOX2 levels, 93% of wild-type mean NANOG levels, and distributions and median expression levels of these proteins were similar to wild-type E14 cells (Fig EV2I). Growth rates were also unaltered (Fig EV2J and Table EV2). Expression levels of pluripotency

markers at the mRNA level (Fig EV1E and Table EV3), cloning efficiency and alkaline phosphatase activity after 1 week of clonal growth were also similar to WT ES cells (Fig EV1F).

We transduced the SNSF cell line with lentiviral constructs allowing dox-inducible overexpression of SOX2-SNAP or YPet-SOX2-delDBD that lacks the SOX2 DNA binding domain. We then monitored SOX2 protein levels over time by luminescence microscopy after dox induction, as well as SOX2-SNAP or YPet-SOX2-delDBD levels by fluorescence microscopy. While YPet-SOX2-delDBD expression did not impact endogenous SOX2 levels, SOX2-SNAP overexpression reduced SOX2 protein levels to 50% of their initial levels after approximately 9 h (Fig 2A) ($P = 1.88 \times 10^{-14}$, Mann–Whitney *U*-test). This time scale is close to the half-life of the SOX2 protein ($8.2 \pm 1$ h (SE) for the half-life of SOX2-SNAP, Fig 1E) and thus raises the possibility of a very rapid arrest of SOX2 protein accumulation. To determine whether SOX2 directly alters *Sox2* mRNA levels, we performed RT–qPCR shortly after overexpression of YPet-SOX2 or YPet-SOX2-delDBD. We found that *Sox2* mRNA levels were decreased already 2 h after dox induction, and were down to 20% of their initial level after 6 h (Fig 2B and Table EV6). This suggests that SOX2 rapidly represses the expression of its own mRNA, in line with earlier studies (Kopp *et al*, 2008; Ormsbee Golden *et al*, 2013). We next aimed to determine how SOX2 overexpression increased OCT4 levels. Surprisingly, *Oct4* mRNA levels were unaffected after 6 h of SOX2 overexpression (Fig 2B and Table EV6). Since SOX2 and OCT4 form heterodimers, we reasoned that SOX2 overexpression could increase OCT4 levels by increasing the stability of the OCT4 protein. We thus measured OCT4 half-life by pulse-labelling OCT4-HALO with the Halo-SiR ligand in the SBROS cell line (as described previously in Alber *et al*, 2018) after 6- to 8-h dox induction of SOX2-SNAP expression. We found the half-life of OCT4 to be increased by 50% (Fig 2C and Table EV7) and to be positively correlated to SOX2-SNAP levels in individual cells (Fig EV2K), suggesting that SOX2 increases OCT4 levels by decreasing its degradation rate. In contrast, SOX2 overexpression does not alter its own half-life, as we have shown previously (Alber *et al*, 2018). To test whether the increase in OCT4 half-life upon SOX2 overexpression depends on the ability of SOX2 to bind DNA, we generated a cell line allowing dox-inducible overexpression of YPet-SOX2-delDBD. We then treated cells with or without dox for 6–8 h and pulse-labelled OCT4-HALO to determine its half-life. Unlike SOX2-SNAP, YPet-SOX2-delDBD overexpression did not increase the half-life of OCT4 (Fig 2C and Table EV8).

We next aimed to determine how endogenous variations in SOX2 and OCT4 levels affect each other's expression levels. To do so, we labelled SBROS cells with SNAP-SiR647 and HaloTag TMR, and sorted cells for either high or low SOX2-SNAP levels, but with the same, intermediate OCT4-HALO expression levels. In order to minimize the effects of cell cycle progression on differences in SOX2 and OCT4 levels, we sorted cells that were in G1 phase based on DNA content (Figs 2D and EV2L). The converse experiment was performed to determine how endogenous OCT4 levels impact SOX2 expression (Figs 2D and EV2L). After sorting, cells were kept either in pluripotency maintenance conditions (N2B27+2iLIF) or in differentiation conditions (N2B27). In cells sorted for SOX2-high or SOX2-low levels, OCT4 levels were increased and decreased 8 h after sorting, respectively (Fig 2E, Table EV9). In contrast, in cells sorted for OCT4-high or OCT4-low levels, SOX2 levels remained unchanged 8 h after sorting (Fig 2F, Table EV9). Note that since

cells typically require at least 4–6 h to re-attach to cell culture dishes after plating, a large fraction of cells were still in G1 phase 8 h after sorting (Fig EV2M). These results suggest that SOX2-high and low cells tend to increase and decrease OCT4 expression levels over time, respectively, and that low amplitude, endogenous variations in SOX2 levels regulate dynamic changes in OCT4 levels.

## Characterization of SOX2 and OCT4 fluctuations

The intercellular variability and the reversion of SOX2 and OCT4 levels towards their mean levels 8 h after sorting for SOX2-high/low

or OCT4-high/low levels (Fig 2E and F, respectively) suggest that these proteins fluctuate over a time scale of hours in individual cells. We thus decided to use our knock-in ES cell lines to directly measure protein expression-level fluctuations at the single-cell level. We monitored absolute SOX2 levels in the SNSF cell line by luminescence microscopy, using a signal calibration approach we reported previously (Mandic et al, 2017). As expected, SOX2 levels doubled over one cell cycle (Figs 2G and EV3A), and SOX2 concentrations calculated after normalization to an inferred nuclear volume (described in Filipczyk et al, 2015) were constant on average (Fig EV3B). In single cells, we found SOX2 concentrations to

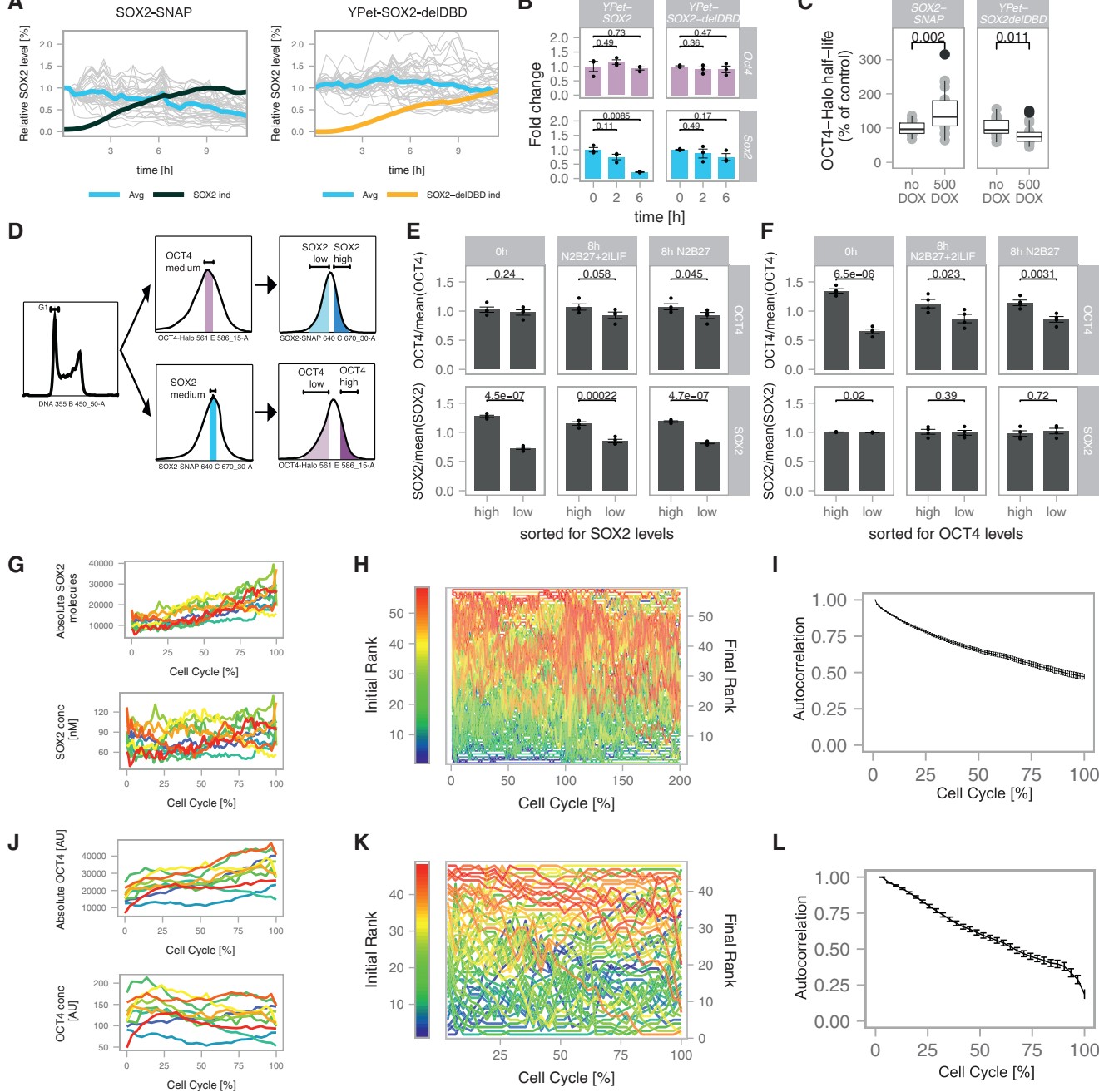

**Figure 2.**

◄

**Figure 2. Cross-regulation and single-cell fluctuations of SOX2 and OCT4.**

A   Endogenous SOX2 levels in single cells (turquoise) normalized to the value at $t = 0$ in SNSF cells upon induction of SOX2-SNAP ($n = 37$, left) or a truncated SOX2 version missing the DNA binding domain (YPET-SOX2-delDBD, $n = 47$, right). Black line: average SOX2-SNAP levels after induction normalized to the maximum level ($n = 22$); Yellow line: average YPET-SOX2-delDBD levels after induction, normalized to the maximum level ($n = 20$).

B   *Sox2* and *Oct4* mRNA levels upon overexpression of YPet-SOX2 or YPET-SOX2-delDBD ($n = 3$). *P*-values: two-sided *t*-test with unequal variance. Error bars: SE.

C   Changes of OCT4 half-life upon SOX2-SNAP or YPET-SOX2-delDBD overexpression ($n = 20$ cells each). Whiskers: minimum and maximum values; box: lower and upper quartiles; solid line: median; solid points: outliers. *P*-values: Mann–Whitney *U*-test. SBROS cells harbouring inducible SOX2-SNAP or YPET-SOX2-delDBD were treated with 0 or 500 ng/ml dox for 6–8 h followed by pulse-labelling of OCT4-HALO and live-cell imaging.

D   Sorting strategy to evaluate the impact of endogenous fluctuations of SOX2 and OCT4 on OCT4 and SOX2 protein levels, respectively.

E   Changes of SOX2 and OCT4 levels after sorting for high and low SOX2 levels in cells expressing intermediate OCT4 levels ($n = 4$). Error bars: SE; *P*-values: one-sided *t*-test with unequal variance. Error bars: SE.

F   Changes of SOX2 and OCT4 levels after sorting for high and low OCT4 levels in cells expressing intermediate SOX2 levels ($n = 4$). *P*-values: one-sided *t*-test with unequal variance. Error bars: SE.

G   Representative traces of the absolute number of SOX2-NLUC molecules (top) and inferred concentration in nM (bottom) in *in silico* synchronized cells ($n = 10$).

H   Single cells were ranked according to their SOX2 expression at $t = 0$ and assigned a colour code for their initial rank from low (blue) to high (red) levels, and changes in ranks over time are shown ($n = 59$).

I   Rank-based autocorrelation function of the SOX2 ranks ($n = 59$). Error bars: SE estimated by bootstrapping.

J   Representative single-cell traces of the integrated intensity of OCT4-HALO in single cells (top) and the corresponding inferred concentration (bottom) in *in silico* synchronized cells ($n = 10$).

K   Single cells were ranked according to their OCT4 expression at t = 0 and assigned a colour code for their initial rank from low (blue) to high (red) levels, and changes in ranks over time are shown ($n = 47$).

L   Rank-based autocorrelation function as in (I) ($n = 47$). Error bars: SE estimated by bootstrapping.

fluctuate over a 2- to 3-fold range (Fig 2G). To measure the time scale of SOX2 concentration fluctuations, individual cells were assigned a rank based on their initial SOX2 level (Fig 2H). We then used a rank-based autocorrelation function to determine the time scale of protein-level fluctuations. This time scale is referred to as the mixing time (Sigal *et al*, 2006) and describes how long it takes for a cell to lose its expression rank and thereby to "mix" its expression level with other cells. Using data from either two or a single full cell cycle, we found SOX2 mixing times on the order of one cell cycle (Figs 2I and EV3C and D). Since < 2-fold changes in SOX2 expression were reported to compromise pluripotency maintenance (Kopp *et al*, 2008), our data suggest that rapid readjustment of SOX2 levels may be required to maintain pluripotency despite fluctuation amplitudes of 2- to 3-fold. We then performed analogous experiments using the SBROS cell line to monitor OCT4-HALO levels over the cell cycle by live fluorescence microscopy, revealing similar fluctuation amplitudes (Figs 2J and EV3E and F), and mixing times (Fig 2K and L) as compared to SOX2. Thus, both SOX2 and OCT4 display 2- to 3-fold, rapid expression-level fluctuations in the pluripotent state.

## SOX2 and OCT4 fluctuations impact NE and ME commitment

We next aimed to determine how endogenous SOX2 levels tune the probability of NE differentiation by monitoring SOX2-NLUC levels and SOX1-P2A-FLUC expression after removal of 2i and LIF (Fig 3A and B). We tracked NLUC and FLUC signals in individual cells over time, and individual cell traces were aligned and normalized for cell cycle duration *in silico*, using linear resampling of the time variable (see Materials and Methods). We then grouped cells according to their expression of FLUC and traced their SOX2 expression levels one cell cycle before, during the cell cycle of FLUC expression onset, and in the subsequent cell cycle. Traces of cells that did not express FLUC were temporally aligned to traces of cells that expressed FLUC, so that both groups of cells were compared at the same average time after which self-renewal signals were withdrawn (Fig 3C). Interestingly, during the cell cycle before Sox1$^+$ cells appeared,

higher SOX2 levels at the beginning and end, but not in the middle of the cell cycle correlated with a higher probability of NE differentiation (Fig 3C, marked by *, see Table EV10 for *P*-values). This suggests that SOX2 levels may play a role in NE commitment at the M-G1 transition. This finding is in line with an earlier study from our laboratory, in which we found that the absence of SOX2 at the M-G1 transition suppresses its ability to enhance neuroectodermal fate commitment (Deluz *et al*, 2016).

We next investigated whether different SOX2 levels at the very onset of differentiation impact NE and ME commitment. To do so, we sorted G1-gated SBROS cells stained with SNAP-SiR647 for low, medium and high SOX2 levels (Figs 3D and EV4A). Cells were then released from self-renewal conditions by seeding in N2B27 medium devoid of 2i and LIF, and 4 days later, NE and ME commitment were assessed by flow cytometry using the SOX1-P2A-eGFP and the BRA-P2A-mCherry reporters as readout. The fraction of eGFP$^+$ cells scaled with initial SOX2 levels (Fig 3E and Table EV11), suggesting that high SOX2 levels at the time of release from self-renewal enhance NE fate commitment. In contrast, SOX2 levels had only a weak impact on ME commitment in these conditions (Fig 3E).

To determine the impact of OCT4 levels at the onset of pluripotency exit on NE and ME commitment, we sorted G1-gated SBROS cells stained with HaloTag TMR in OCT4-low and OCT4-high subpopulations (Figs 3D and EV4B) and cultured them for 4 days in the absence of LIF and 2i. Surprisingly, we found a large difference in NE and ME commitment between these populations (Fig 3F and Table EV11), even though average OCT4 levels differed by < 2-fold between them (Fig EV4B). We next aimed to explore the potential causal relationship between high OCT4 levels and increased NE/ME commitment. To do so, we generated a cell line allowing for inducible expression of SNAP-OCT4 in the SBR background and treated it with dox for 12 h to overexpress OCT4 for a brief period of time, thereby mimicking the timescales of endogenous OCT4 fluctuations (note that we previously demonstrated the functionality of the SNAP-OCT4 fusion protein (Deluz *et al*, 2016)). Subsequently, cells were sorted for G1 phase and differentiated for 4 days in N2B27 medium without 2i and LIF

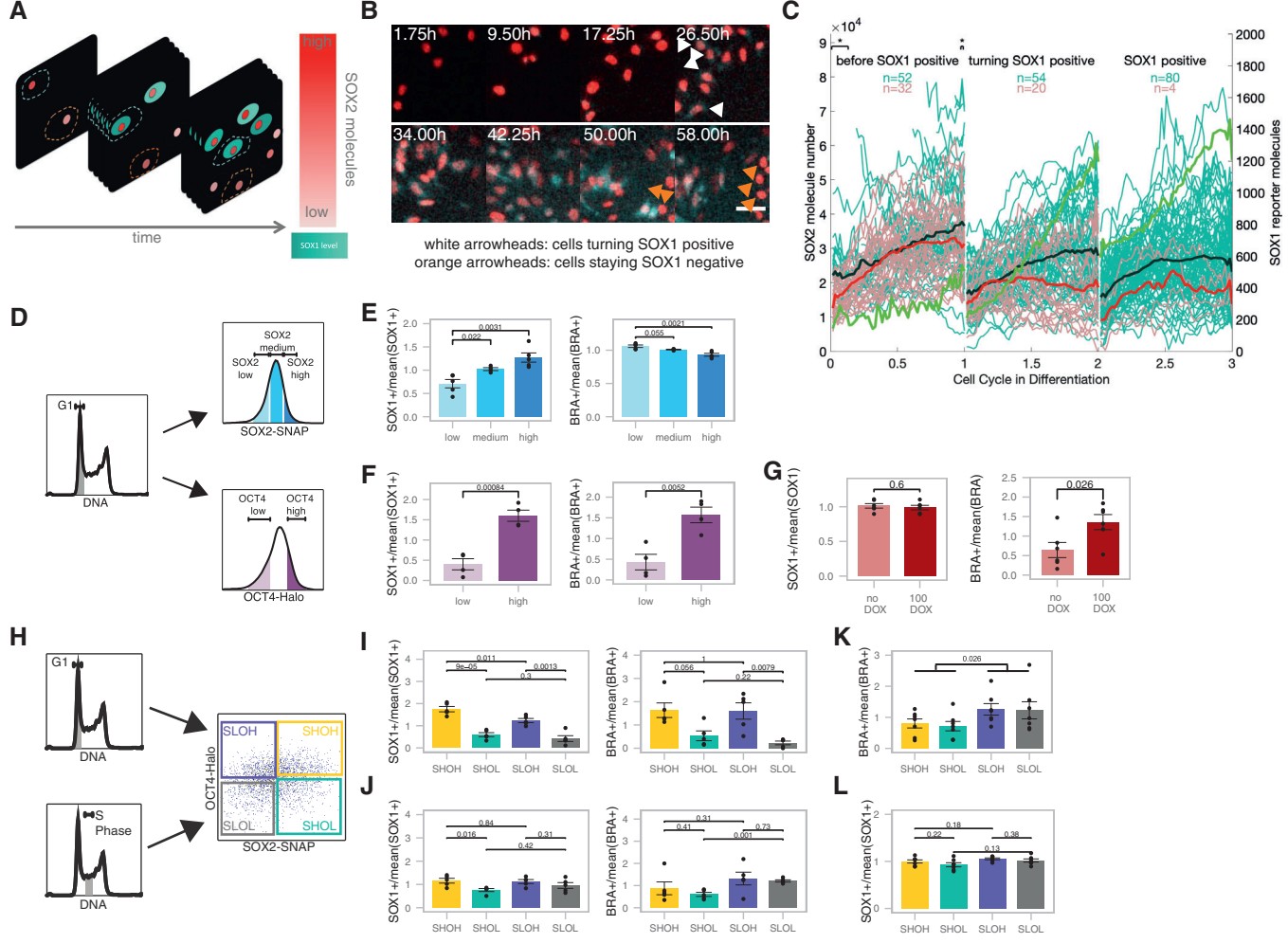

**Figure 3. Endogenous SOX2- and OCT4-level fluctuations bias differentiation.**

A   Scheme of single-cell tracking for SNSF cells turning SOX1-positive (turquoise cytoplasm) or remaining SOX1-negative throughout the experiment. Turquoise dashed contour: measured region for cells turning SOX1 positive; Red dashed contour: measured region for cells staying SOX1 negative.

B   Snapshot from luminescence movie of SNSF cells in differentiation. White arrowheads: cells starting to express SOX1-P2A-FLUC; Orange arrowheads: cells staying SOX1-negative throughout the experiment. Red: SOX2-NLUC, turquoise: Sox1-P2A-FLUC. Scale bar: 50 μm.

C   Single-cell traces of SOX2 levels in differentiating cells. Turquoise traces: cells becoming SOX1-positive in the middle cell cycle, population average is shown in bold black; Red traces: cells remaining SOX1-negative throughout the experiment, population average is shown in bold; Green line: population average FLUC signal in cells that become SOX1-positive. *P < 0.05 determined by two-sided *t*-test with unequal variance. See Table EV10 for *P*-values.

D   Strategies to sort SBROS cell populations with different SOX2 and OCT4 levels.

E   Differentiation outcome for cells sorted for different SOX2 levels (*n* = 5 for low and high, *n* = 4 for medium). *P*-values: two-sided *t*-test with unequal variance. Error bars: SE.

F   Differentiation outcome for cells sorted for different OCT4 levels (*n* = 4). *P*-values: two-sided *t*-test with unequal variance. Error bars: SE.

G   Differentiation outcome of uninduced cells or cells induced 12 h for OCT4-SNAP overexpression before sorting in G1 phase (*n* = 6). *P*-values: two-sided *t*-test with unequal variance. Error bars: SE.

H   Sorting strategy for SBROS cells in G1 or S phase and into the four following subpopulations: SOX2 high & OCT4 high (SHOH), SOX2 high & OCT4 low (SHOL), SOX2 low & OCT4 high (SLOH) or SOX2 low & OCT4 low (SLOL).

I   Differentiation outcome for cells sorted in G1 phase (*n* = 5). *P*-values: two-sided *t*-test with unequal variance (SOX1) and Mann–Whitney *U*-test (BRA). Error bars: SE.

J   Differentiation outcome for cells sorted in S phase (*n* = 5). *P*-values: Mann–Whitney *U*-test (SOX1) and two-sided *t*-test with unequal variance (BRA). Error bars: SE.

K   Differentiation outcome for cells sorted in G1 phase differentiated in the presence of 3 μM CHIR (*n* = 7). *P*-values: two-sided *t*-test with unequal variance. Error bars: SE.

L   Differentiation outcome for cells sorted in G1 phase differentiated in the presence of 1 μM SB431542 and 25 ng/ml bFGF (*n* = 7). *P*-values: two-sided *t*-test with unequal variance. Error bars: SE.

(Figs 3G and EV4C). Strikingly, this led to a ~2-fold increase in the fraction of mCherry[+] cells, suggesting that even a brief increase of OCT4 levels prior to self-renewal release enhances ME commitment. However, NE commitment was not enhanced as the fraction of eGFP[+] cells remained unchanged (Figs 3G and EV4C

and Table EV12). We reasoned that this discrepancy could be caused by opposing roles of high OCT4 levels before and after removal of pluripotency signals. To investigate the impact of OCT4 overexpression after release of pluripotency signals, we treated SNAP-OCT4 cells with dox for 4 days after removal of 2i

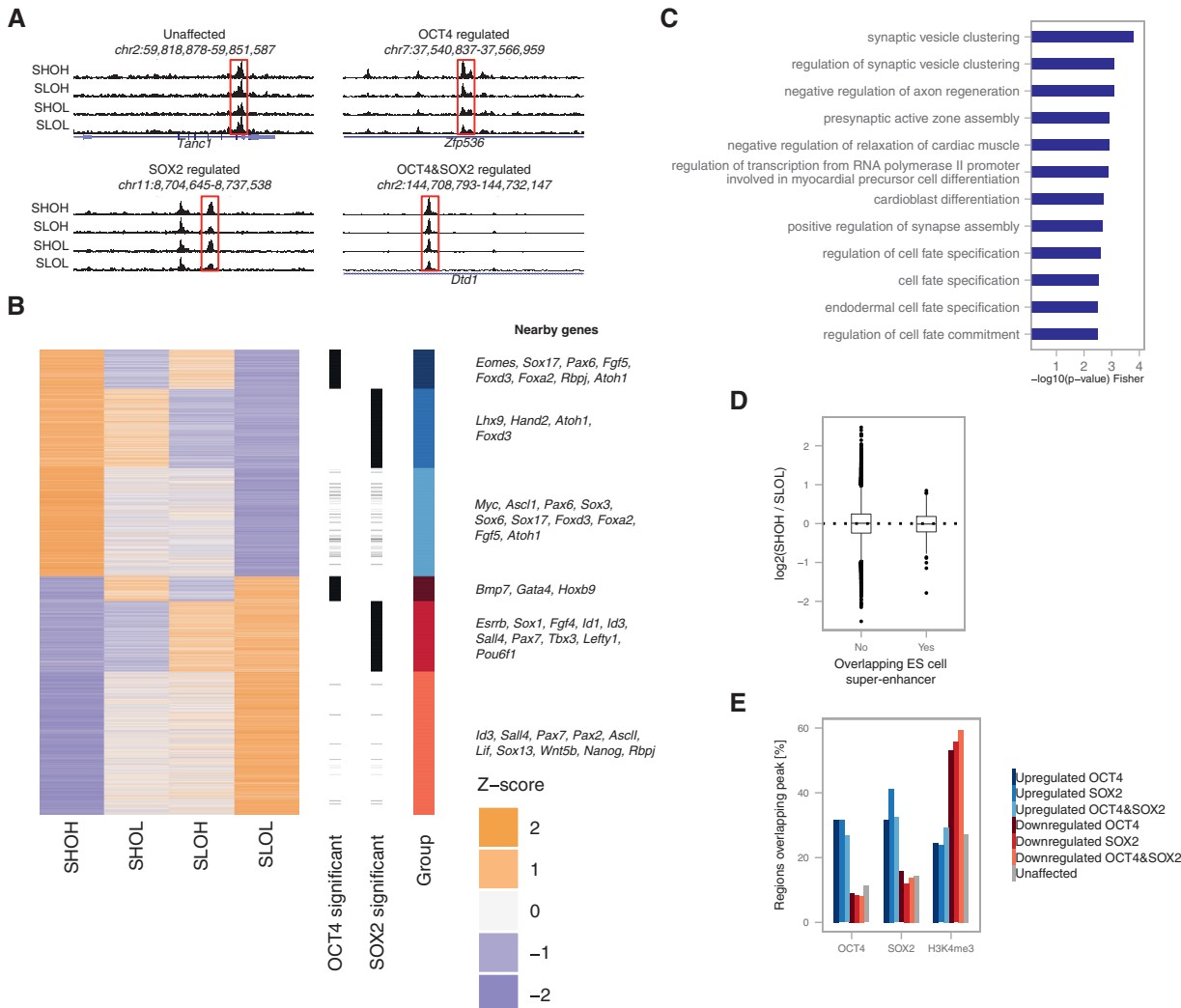

**Figure 4. SOX2- and OCT4-dependent changes in chromatin accessibility.**

A  Example tracks of the ATAC-seq signal in SHOH, SHOL, SLOH and SLOL populations for loci where chromatin accessibility is unaffected, affected by OCT4 levels, SOX2 levels or affected by both.

B  Heatmap of relative chromatin accessibility in SHOH/SHOL/SLOH/SLOL samples for all loci with significantly altered accessibility, with examples of genes close to affected regulatory regions. Groups are defined by their significance (OH versus OL/SH versus SL/SHOH versus SLOL) and fold-change direction.

C  Gene Ontology (GO) term enrichment analysis for genes close to regions with higher accessibility in OCT4-high cells (upregulated OCT4).

D  Chromatin accessibility ratio of SHOH versus SLOL cells in loci overlapping ES cell super-enhancer regions or not. Whiskers: minimum and maximum values; box: lower and upper quartiles; solid line: median; solid points: outliers.

E  Percentage of regions in each group showing an overlap with OCT4, SOX2 and H3K4me3 ChIP-seq peaks.

and LIF in order to overexpress OCT4 during the differentiation process. While this also led to a similar increase in ME differentiation as obtained with a short dox treatment, NE commitment was strongly inhibited (Fig EV4D and E, Table EV13). In summary, our data suggest that high OCT4 levels throughout differentiation inhibit NE commitment (in line with Thomson *et al*, 2011), in contrast to high endogenous OCT4 levels before the onset of differentiation that enhance NE commitment. Finally, providing a short pulse of dox induction in the SNAP-OCT4 inducible SBR cell line before differentiation yields an intermediate NE differentiation outcome, which can be explained by reaching supraphysiological OCT4 levels both before and at the onset of differentiation.

We next measured the combinatorial impact of SOX2 and OCT4 on differentiation by sorting G1-gated SBROS cells into four different subpopulations (Figs 3H and EV4F) followed by 4 days of differentiation (Figs 3I and EV4G and Table EV11). As expected, SOX2-high/OCT4-high (SHOH) cells were the most efficient to differentiate towards NE and ME, SOX2-low/OCT4-high (SLOH) cells were less capable to differentiate towards NE as compared to SHOH, and OCT4-low (SHOL and SLOL) populations were strongly impaired in differentiating towards both fates. Overall, OCT4 levels had a larger impact than SOX2 levels on NE and ME differentiation. Furthermore, as OCT4-high cell populations had relatively similar differentiation efficiencies regardless of SOX2 levels, we conclude that the

**Table 1. Sequences of guide RNAs used to generate knock-in cell lines.**

| gRNA targets | gRNA sequence (5′–3′) |
|---|---|
| Oct4 | GACTGAGGCACCAGCCCTCCC |
| Sox2 | GCAGCCCTCACATGTGCGACA |
| Sox1 | GACGCACATCTAGCGCCGCG |

OCT4/SOX2 ratio is not a strong determining factor in differentiation outcome. To ask whether endogenous SOX2- and OCT4-level variability impacts NE and ME commitment during later cell cycle stages, we performed the same experiments in S phase-gated cells. We found the impact of SOX2 on NE commitment and the impact of OCT4 on both NE and ME commitment to be decreased (Figs 3J and EV4H and Table EV11). Note that the differentiation efficiency in our conditions was not lower in S phase-sorted as compared to G1 phase-sorted cells, in contrast to previous studies (Coronado et al, 2013; Pauklin & Vallier, 2013). In fact, while ME differentiation of S phase-sorted cells was comparable to G1 phase-gated cells, NE differentiation was increased (Fig EV4I). These discrepancies likely result from differences in differentiation protocols and/or endpoint readouts. Regardless, our data suggest that differentiation to NE and ME depends on the levels of OCT4 and SOX2 in G1 but not in S phase, which in the case of SOX2 is in line with its reported function at the Mitosis-G1 transition in NE induction (Deluz et al, 2016).

We next aimed at determining whether SOX2 and OCT4 levels also impact cell fate commitment upon directed differentiation. To do so, we cultured G1-sorted SHOH, SHOL, SLOH and SLOL subpopulations in N2B27 supplemented with 3 μM CHIR99021 during 3 days to drive ME differentiation (Turner et al, 2014). Importantly, only very low percentages of SOX1-positive cells were detected under this regime (Fig EV4J and Table EV11). Surprisingly, we found that OCT4-high and OCT4-low cells displayed a similar potential to differentiate towards ME (Fig 3K and Table EV11). In contrast, SOX2-low cells were almost twofold more efficient than SOX2-high cells to differentiate towards ME (Fig 3K). This suggests that the sustained activation of the Wnt pathway overcomes the influence of OCT4 in priming ME differentiation, and that in this context SOX2 expression levels become determining for ME differentiation efficiency. We next differentiated G1-sorted SHOH, SHOL, SLOH and SLOL cells in the presence of 1 μM SB431542 and 25 ng/ml bFGF to direct cells towards neuroectoderm. This increased the proportion of SOX1-positive cells and decreased the proportion of

BRA-positive cells to about 0.5% (Fig EV4J and Table EV11). In these conditions, there was no significant difference in the differentiation outcome of SHOH, SHOL, SLOH and SLOL cells (Fig 3L). Taken together, these results suggest that the impact of endogenous OCT4 and SOX2 fluctuations on ES cell differentiation is highly context-dependent.

**OCT4-high cells open differentiation enhancers**

We then aimed to identify the molecular mechanisms by which small and transient endogenous fluctuations of SOX2 and OCT4 can result in major biases in differentiation potential. As SOX2 and OCT4 were shown to regulate chromatin accessibility (King & Klose, 2017; Raccaud et al, 2019), we reasoned that small changes in their expression level could prime cells for different fates by altering the chromatin accessibility landscape. We thus performed ATAC-seq in G1-sorted SHOH, SHOL, SLOH and SLOL cells. We quantified the fraction of reads in 81,132 open regions identified by peak calling and found no overall differences between conditions (Fig EV5A). We compared the accessibility of each open region in high versus low conditions for SOX2 and OCT4 as well as for SHOH versus SLOL. 3,914 loci (4.8%) were significantly up- or downregulated (FDR < 10%) upon changes in OCT4 alone (538 loci), SOX2 alone (1,259 loci), or SOX2 and OCT4 together, i.e. SHOH versus SLOL (2,117 loci). We grouped these loci into OCT4-regulated, SOX2-regulated and co-regulated loci that were either upregulated (more open in high cells) or downregulated (more open in low cells) (Figs 4A and B, and EV5B and C). Loci in all groups were close to differentiation-associated genes (Fig 4B) but those upregulated by OCT4 alone were the most enriched for genes involved in differentiation processes (Figs 4C and EV5D–H). Using a subset of OCT4-upregulated loci that overlap OCT4 peaks and with a more stringent false discovery rate (< 5%) also revealed enrichment for differentiation terms (Fig EV5I), showing that OCT4 binds to regulatory elements near differentiation genes that are more accessible in cells with higher OCT4 levels. In contrast, loci overlapping pluripotency-associated super-enhancers were unaffected by SOX2 and OCT4 levels (Fig 4D). Loci in which accessibility was positively correlated to SOX2 and OCT4 levels were enriched for SOX2 and OCT4 binding, while those that were negatively correlated showed less overlap with SOX2 and OCT4 ChIP-seq peaks and were enriched for H3K4me3 (Fig 4E), marking active promoters and transcription start sites, and close to promoters (Fig EV5J). Note that while we used SOX2 and OCT4 peaks

**Table 2. Primers for knock-in verification.**

| Name | Sequence (5′–3′) | Target |
|---|---|---|
| Sox1-KI_F2 | GTGCCCCTGACGCACAT | In 5′ HA |
| Sox1-KI_R2 | CGCTGTGTGCCTCCTCTG | In 3′ HA |
| Seq_Sox2_KI_fw | AGGTGCCGGAGCCCG | In 5′ HA |
| Sox2_verif_rv_3′ | GCATGCTAGCCACAAAGAAA | Downstream of 3′ HA |
| Oct4_KI_verif_fw | GCTCCTCCACCCACCC | Within Oct4 |
| (sf)GFP_verif_fw | CTCGGCATGGACGAGC | In sfGFP |
| Seqout_IRES_rv | AGACAGGGCCAGGTTTCC | In IRES |
| COct4_5′_500bp_f | GATCGTCGACTAGCACAATCCCTTAGCGGT | Upstream of 5′ HA |

**Table 3. Antibodies used for Western blotting.**

| Target | Species | Dilution | Supplier |
|---|---|---|---|
| Primary antibodies | | | |
| αSOX2 | Rabbit | 1:250 | Invitrogen 48-1400 |
| αOCT4 | Mouse | 1:200 | Santa Cruz sc5279 |
| Secondary antibodies | | | |
| αMouse-IgG-HRP | Goat | 1:10,000 | Promega W402B |
| αRabbit-IgG-HRP | Goat | 1:10,000 | Promega W401B |

**Table 4. Primers used for qPCR.**

| Primer name | Primer sequence (5′–3′) |
|---|---|
| Nanog_qPCR_f | AACCAAAGGATGAAGTGCAAGCGG |
| Nanog_qPCR_r | TCCAAGTTGGGTTGGTCCAAGTCT |
| Esrrb_qPCR_f | GCACCTGGGCTCTAGTTGC |
| Esrrb_qPCR_r | TACAGTCCTCGTAGCTCTTGC |
| Sox2_qPCR_f | GCACATGAACGGCTGGAGCAACG |
| Sox2_qPCR_r | TGCTGCGAGTAGGACATGCTGTAGG |
| Oct4_qPCR_f | GAGGAGTCCCAGGACATGAA |
| Oct4_qPCR_r | AGATGGTGGTCTGGCTGAAC |
| Klf4_qPCR_f | CGAACTCACACAGGCGAGAA |
| Klf4_qPCR_r | CGGAGCGGGCGAATTT |
| Rps9_qPCR_f | TTGTCGCAAAACCTATGTGACC |
| Rps9_qPCR_r | GCCGCCTTACGGATCTTGG |
| Zfp42_qPCR_f | CCCTCGACAGACTGACCCTAA |
| Zfp42_qPCR_r | TCGGGGCTAATCTCACTTTCAT |

**Table 5. Antibodies used for immunofluorescence staining.**

| Target | Species | Dilution | Supplier |
|---|---|---|---|
| Primary antibodies | | | |
| αSOX2 | Rabbit | 1:200 | Invitrogen Cat#48-1400 |
| αOCT4 | Mouse | 1:50 | Cell Signaling Cat#75463 |
| αOCT4 | Mouse | 1:300 | Santa Cruz Cat#sc5279 |
| αNANOG | Rabbit | 1:500 | Abcam Cat#ab80892 |
| Secondary antibodies | | | |
| αMouse-IgG-555 | Donkey | 1:1,000 | Thermo Fisher Cat#A31570 |
| αMouse-IgG-488 | Goat | 1:1,000 | Thermo Fisher Cat#A11001 |
| αMouse-IgG-647 | Goat | 1:1,000 | Thermo Fisher Cat#A21235 |
| αRabbit-IgG-647 | Chicken | 1:1,000 | Thermo Fisher Cat#A21443 |

mark H3K4me1 (Table EV14). One of the factors with enriched overlap in the upregulated regions was NANOG (Table EV14). Accordingly, NANOG binding was strongly enriched at loci whose accessibility was upregulated in OCT4-high or SOX2-high cells compared to regions with downregulated accessibility (Fig EV5N). This suggests that NANOG may act as a binding partner to regulate chromatin accessibility at these loci, in line with a recent study (Heurtier *et al*, 2019). Loci with increased accessibility in OCT4-high cells lose accessibility upon rapid OCT4 depletion (data from King & Klose, 2017), in line with OCT4 directly regulating accessibility at these sites (Fig EV5O) even though we cannot exclude that some loci may be regulated by indirect mechanisms. Taken together, our data suggest that fluctuations of endogenous SOX2 and OCT4 lead to temporal changes in chromatin accessibility, and that high OCT4 levels result in the opening of differentiation-associated enhancers.

# Discussion

While gene expression fluctuations are increasingly being recognized as an important source of protein-level variability in single cells, how these impact cellular functions remains largely unclear. Here we show that endogenous fluctuations in levels of pluripotency regulators have a major impact on ES cell differentiation potential. While Nanog displays prolonged, large amplitude fluctuations that alter ES cell differentiation potential in serum + LIF (Kalmar *et al*, 2009), these are caused by transitions between naïve and primed ES cells (Filipczyk *et al*, 2015) and thus reflect fluctuations between different phenotypic states. In contrast, ES cells maintained in a naïve state display small amplitude, transient fluctuations of SOX2 and OCT4, which nonetheless bias differentiation potential.

Near-complete OCT4 depletion was shown to lead to upregulation of trophectoderm markers (Niwa *et al*, 2000). In contrast, less dramatic decreases in WT OCT4 levels lead to enhanced pluripotent self-renewal, as well as compromised differentiation towards ME and NE (Karwacki-Neisius *et al*, 2013; Radzisheuskaya *et al*, 2013), in line with our findings. Conversely, constant OCT4 overexpression was shown to lead to enhanced differentiation towards the mesoderm lineage (Niwa *et al*, 2000; Thomson *et al*, 2011). This is also in line with our findings that OCT4-high cells are more prone to

from two ChIP-seq datasets generated in the same conditions for meaningful comparison, which yielded ~30% of overlap of upregulated loci, including peaks from two additional OCT4 ChIP-seq datasets (with no comparable SOX2 data) increased this number to ~70% (Fig EV5K), indicating that most of these loci are bound by OCT4 and possibly also by SOX2. The OCT motif was most enriched near loci upregulated in OCT4-high cells, and the SOX motif was most enriched near loci upregulated in SOX2-high cells (Fig EV5L). However, the fraction of sites overlapping these canonical motifs was low, suggesting that OCT4 and SOX2 may bind indirectly with other partners. To perform a search for binding partners in these regions, we analysed the overlap with peaks from 3,724 ChIP-seq datasets in ES cells extracted from cistromeDB (Mei *et al*, 2017). We trained a random forest model to categorize loci in the different groups (see Materials and Methods), including accessible regions that were unaffected, based on the peaks overlapping the regions. While the model performed poorly in predicting the group a region belonged to based on the peaks a region overlapped, it performed better at distinguishing between upregulated, downregulated and unaffected loci (Fig EV5M). We noticed that many of the top classifiers enriched in downregulated loci included promoter and gene body signatures such as Cdk9, Pol2 and H3K79me2, consistent with these loci being enriched near genes, while upregulated loci were enriched for the enhancer

differentiate towards ME than OCT4-low cells. The enhancement of NE commitment by elevated SOX2 levels that we described here is also in line with previous studies (Zhao *et al*, 2004; Thomson *et al*, 2011; Wang *et al*, 2012), but here we show that enhanced NE is not only caused by SOX2 overexpression throughout differentiation, but that high endogenous SOX2 levels at the onset of differentiation are sufficient to significantly bias ES cells towards the NE lineage. At first sight, the enhancement of NE commitment by high endogenous OCT4 levels seems to contradict previous work (Thomson *et al*, 2011). However, this study used either OCT4 overexpression or measured endogenous OCT4 levels at fixed time points, throughout directed NE or ME differentiation. While we confirmed that OCT4 overexpression during differentiation suppresses NE differentiation, we also found that elevated OCT4 levels at the time self-renewal signals are removed increases NE commitment. Taken together, this suggests that while high OCT4 levels consistently increase ME commitment both before and during differentiation, high OCT4 levels enhance NE differentiation only at the onset of self-renewal signals removal. We also found that in the context of directed ME and NE differentiation, the impact of endogenous OCT4 and SOX2 on cell fate commitment becomes much weaker. Therefore, the modulation of signalling pathways can partially or completely buffer the impact of endogenous fluctuations of OCT4 and SOX2 on ES cell differentiation.

The large impact of small amplitude, transient OCT4 fluctuations on ES cell differentiation in the absence of strong signalling cues suggests a sensitive and rapid downstream mechanism modulating cell responsiveness to differentiation. While changes in the chromatin accessibility landscape as a function of OCT4 levels offer an explanation for these observations, the reason for differential responses of pluripotency regulatory elements as compared with differentiation-related enhancers is unclear. Using thousands of ChIP-seq datasets in ES cells, we could train a model to predict, with relatively good accuracy, regions that were upregulated or downregulated in accessibility upon high levels of OCT4 or SOX2 (Fig EV5M). We identified NANOG binding as enriched in upregulated regions, consistent with a recent report on the impact of NANOG depletion on OCT4/SOX2 binding and accessibility (Heurtier *et al*, 2019). However, we could not find features allowing to predict whether loci are more dependent on OCT4 versus SOX2. The potential role for cooperativity with other proteins or differential affinity of OCT4/SOX2 binding sites will require further investigation. We observed widespread SOX2-dependent accessibility changes, including at many differentiation enhancer-associated loci. However, loci with increased accessibility in SOX2-high cells were less enriched for differentiation-associated enhancers than in OCT4-high cells (Fig EV5D), which is in line with the weaker effects of SOX2 levels on differentiation. Finally, the fact that cells are more sensitive to SOX2- and OCT4-level variability in G1 phase than S phase raises the possibility that these transcription factors mainly act shortly after mitosis to re-open closed enhancer regions, in line with their reported pioneer transcription factor activity (Soufi *et al*, 2015) and their essential function in cell fate decisions at the mitosis-G1 transition (Deluz *et al*, 2016; Liu *et al*, 2017b).

# Materials and Methods

### Reagents and Tools table

| Reagent/Resource | Reference or source | Identifier or catalog number |
|---|---|---|
| **Experimental models** | | |
| SBR | Deluz *et al* (2016) | |
| SNSF | This study | |
| Calibration cells | Mandic *et al* (2017) | |
| 2TS22C | Masui *et al* (2007) | |
| Zhbtc4 | Niwa *et al* (2000) | |
| E14 | Trono lab, EPFL | |
| CGR8 | Sigma | |
| HEK293T | ATCC | |
| **Recombinant DNA** | | |
| pKI Sox2-Nluc-loxP-P2A-Puro-sfGFP-loxP | This study | |
| pKI Sox2-Nluc-loxP-P2A-Bsd-eGFP-loxP | This study | |
| pKI Sox2-SNAP-IRES-Hygro | This study | |
| pKI Oct4-HALO-IRES-Bsd | This study, Addgene | #115689 |
| pKI Sox1-P2A-FLuc-loxP-pGK-Hygro-loxP | This study | |
| pX330 | Cong *et al* (2013), Addgene | #42230 |
| pX335 | Cong *et al* (2013), Addgene | #42335 |
| pX330 Sox2 | This study | |

**Reagents and Tools table**   (continued)

| Reagent/Resource | Reference or source | Identifier or catalog number |
|---|---|---|
| pX330 Oct4 | This study | |
| pX335 Sox1 | This study | |
| pLV TRE3G-Ypet-Sox2 | Deluz *et al* (2016) | |
| pLV TRE3G-Ypet-Oct4 | Deluz *et al* (2016) | |
| pLV TRE3G-SOX2-SNAP | Deluz *et al* (2016) | |
| pLV TRE3G-YPet-Sox2-delDBD | Deluz *et al* (2016) | |
| pLV pGK-rtTA3G-IRES-Hygro | Mandic *et al* (2017) | |
| pLV pGK-rtTA3G-IRES-Bsd | Deluz *et al* (2016) | |
| pLV-pGK-Cre | Deluz *et al* (2016) | |
| pLV pGK-Nluc-Fluc-NLS-P2A-H2B-mCherry-IRES-Hygro | Mandic *et al* (2017), Addgene | #115684 |
| **Antibodies** | | |
| Rabbit anti-SOX2 | Invitrogen | 48-1400 |
| Goat anti-SOX2 | Santa Cruz | sc-17320 |
| Mouse anti-OCT4 | Santa Cruz | sc-5279 |
| Mouse anti-OCT4 | Cell Signaling | 75463S |
| Rabbit anti-OCT4 | Cell Signaling | 5677S |
| Rabbit anti-NANOG | Abcam | ab80892 |
| Donkey anti-Mouse-IgG-555 | ThermoFisher | A31570 |
| Goat anti-Mouse-IgG-488 | ThermoFisher | A11001 |
| Goat anti-Mouse-IgG-647 | ThermoFisher | A21235 |
| Chicken anti-Rabbit-IgG-647 | ThermoFisher | A21443 |
| Goat anti-Mouse-IgG-HRP | Promega | W402B |
| Goat anti-Rabbit-IgG-HRP | Promega | W401B |
| **Oligonucleotides** | | |
| Sox1-KI_F2 | This study | Table 2 |
| Sox1-KI_R2 | This study | Table 2 |
| Seq_Sox2_KI_fw | This study | Table 2 |
| Sox2_verif_rv_3′ | This study | Table 2 |
| Oct4_KI_verif_fw | This study | Table 2 |
| (sf)GFP_verif_fw | This study | Table 2 |
| Seqout_IRES_rv | This study | Table 2 |
| COct4_5′_500bp_f | This study | Table 2 |
| Nanog_qPCR_f | This study | Table 4 |
| Nanog_qPCR_r | This study | Table 4 |
| Esrrb_qPCR_f | This study | Table 4 |
| Esrrb_qPCR_r | This study | Table 4 |
| Sox2_qPCR_f | This study | Table 4 |
| Sox2_qPCR_r | This study | Table 4 |
| Oct4_qPCR_f | This study | Table 4 |
| Oct4_qPCR_r | This study | Table 4 |
| Klf4_qPCR_f | This study | Table 4 |
| Klf4_qPCR_r | This study | Table 4 |
| Rps9_qPCR_f | This study | Table 4 |
| Rps9_qPCR_r | This study | Table 4 |
| Zfp42_qPCR_f | This study | Table 4 |
| Zfp42_qPCR_r | This study | Table 4 |

## Methods and Protocols

### Cell culture

The SBR (Deluz *et al*, 2016) and SBROS cell lines were generated from CGR8 ES cells (Sigma, Cat#07032901-1VL), and the E14 cell line (kindly provided by Didier Trono, EPFL) was used for all ES cell experiments. Cells were routinely cultured on dishes coated with 0.1% gelatin type B (Sigma, Cat#G9391-100G), in GMEM (Sigma, Cat#G5154-500ML) supplemented with 10% ES cell-qualified foetal bovine serum (Gibco, Cat#16141-079), nonessential amino acids (Gibco, Cat#11140-050), 2 mM L-glutamine (Gibco, Cat#25030-024), sodium pyruvate (Sigma, Cat#S8636-100ML), 100 μM 2-mercaptoethanol (Sigma, Cat#63689-25ML-F), penicillin and streptomycin (BioConcept, Cat#4-01F00-H), homemade leukaemia inhibitory factor (LIF), CHIR99021 (Merck, Cat#361559-5MG) at 3 μM and PD184352 (Sigma PZ0181-25MG) at 0.8 μM. Cells were passaged by trypsinization (Sigma, Cat#T4049-100ML) every 2–3 days at a ratio of 1:10.

For imaging experiments, ES cells were cultured on dishes coated with 5 μg/ml E-Cadherin, in N2B27 medium supplemented with LIF, CHIR99021 at 3 μM and PD184352 at 0.8 μM (N2B27+2iLIF). E-Cadherin coating was performed as previously described (Nagaoka *et al*, 2006). Briefly, 5 μg/ml E-Cadherin (R&D, Cat#8875-EC or Cat#748-EC) in PBS (with $Ca^{2+}$ and $Mg^{2+}$; Amimed, Cat#3-05K00-I) was added to the culture vessel and incubated for 90 min at 37°C. Just before seeding, the E-Cadherin solution was removed, the surface of the vessel rinsed once with PBS and filled with the appropriate cell culture medium.

N2B27 medium was prepared by 1:1 mixing of DMEM/F12 (Gibco, Cat#11320-033) + N2 supplement (Gibco, Cat#17502-001) medium with Neurobasal (Gibco, Cat#21103-049) + B27 supplement (Gibco, Cat#17504-001) medium, supplemented with penicillin (1,000 IU/ml) and streptomycin (1,000 mg/ml), 2 mM L-Glutamine and 0.1 mM 2-mercaptoethanol.

HEK293T cells were cultured in DMEM (Gibco, Cat#41966-029) supplemented with 10% foetal bovine serum (Gibco, Cat#10270-106), penicillin and streptomycin and passaged every 2 days at a ratio of 1:8.

For the selection of transduced and transfected cells, the following antibiotic concentrations were used: 8 μg/ml of Blasticidin (Gibco A11139-03), 2 μg/ml of Puromycin (Gibco A11138-03) and 200 μg/ml of Hygromycin B (Invitrogen 10687010). ES cells were transfected using X-tremeGENE 9 transfection reagent (Sigma, Cat#06 365 809 001).

### Generation of knock-in cell lines

The SBROS and SNSF cell lines were generated using CRISPR/Cas9-mediated homology-directed repair (HDR). The repair templates were designed to contain a knock-in cassette flanked by homology arms (HAs) with the target sequence missing the endogenous STOP codon. Guide RNA sequences were designed to overlap with the endogenous STOP codon, and the repair templates contain mutations in the PAM sequence, thus ensuring that the repair plasmids are not cut (Fig EV1A).

The knock-in cassette (between the HAs) contains the coding sequence for the tag in frame with the protein of interest and a selection marker. For the SOX2 knock-ins, the cassette consists of SNAP-IRES-Hygro (SBROS), or NLuc-loxP-P2A-Puro-sfGFP-loxP and NLuc-loxP-P2A-Bsd-sfGFP-loxP (SNSF). The knock-in cassette for

OCT4 consists of a linker (WRAASRLTS)-Halo-IRES-Blasticidin (SBROS). The Sox1 knock-in cassette consists of P2A-FLuc-Stop-loxP-pGK-Hygro-Stop-loxP.

Guide RNAs targeting the *Oct4*, *Sox2* and *Sox1* loci were designed using the Zhang Lab toolbox (www.genome-engineering.org/crispr) and cloned into the pX330 vector (*Sox2* and *Oct4*), expressing Cas9 and the guide RNA, or pX335 (*Sox1*), expressing Cas9n and the guide RNA (Cong *et al*, 2013). The guide RNA sequences are listed in Table 1.

To generate the SBROS cell line, SBR cells (Deluz *et al*, 2016) were transfected with pX330-Sox2 and pKI-SOX2-SNAP-IRES-Hygro at a 1:3 ratio. After 2 days, selection was started with Hygromycin B. After 11 days of selection, cells were stained with 12 nM SNAP-SiR 647, and single cells were sorted for high SNAP expression into 96-well plates and grown out. One clone identified as homozygously targeted by PCR on genomic DNA (Fig EV1B) was further validated by Western blot (Fig EV1C) and used to knock-in a HaloTag at the C-terminus of OCT4. To do so, this clone was transfected with pX330-Pou5f1 and pKI-OCT4-HALO-IRES-Bsd at a 1:3 ratio followed by Blasticidin selection 2 days after transfection. Single colonies were then picked manually and grown out, and one clone in which one allele was targeted as indicated by PCR on genomic DNA (Fig EV1B) was further analysed by Western blot (Fig EV1C).

To generate the SNSF cell line, E14 cells were co-transfected with pX330-Sox2 and pKI-Sox2-NLuc-loxP-P2A-Puro-sfGFP-loxP at a 1:3 ratio. After 2 days, selection with Puromycin was started. After 6 days of selection, cells were co-transfected with pX330-Sox2 and pKI-Sox2-NLuc-loxP-P2A-Bsd-eGFP-loxP, and selection with Blasticidin was initiated 2 days later. A homozygous Blasticidin-resistant clone was identified by PCR on genomic DNA and subsequently recombined by transient transfection of a plasmid-expressing Cre recombinase. Successful excision of the selection cassette was confirmed by PCR on genomic DNA (Fig EV1B). This resulting intermediate cell line (Sox2-NLuc) was co-transfected with pX335-Sox1 and pKI-Sox2-P2A-FLuc-loxP-pGK-Hygro-loxP, and selection with Hygromycin B was started 2 days later. Single clones were picked manually 10 days later, and the knock-in was confirmed using PCR on genomic DNA (Fig EV1B). The fusion of NLUC to SOX2 was confirmed by Western blotting (Fig EV1C).

All knock-in and corresponding wild-type alleles were verified by Sanger sequencing of the PCR products. All sequences were preserved except for the presence of a single nucleotide insertion in the 3′UTR of the wild-type *Oct4* allele of the SBROS cell line.

### Lentiviral vector production and generation of stable cell lines

Lentiviral vectors were produced by calcium phosphate co-transfection of HEK293T cells with the envelope (PAX2), packaging (MD2G) and lentiviral construct of interest. The viral vectors were concentrated 120-fold by ultracentrifugation as described previously (Suter *et al*, 2006). Stable cell lines were generated by transducing 50,000 cells per well of a 24-well plate with 50 μl of concentrated lentiviral vector particles. Antibiotic selection was started 48–72 h later and maintained throughout passaging.

### Alkaline phosphatase assays

ES cells were plated at 400 cells per well of a gelatinated 6-well plate in medium with 10% serum, 2i and LIF. The medium was

exchanged every 2–3 days, and after 7 days, alkaline phosphatase staining was performed following the manufacturer's instruction (Sigma, Cat#86R-1KT).

## Pluripotency rescue colony scoring

Pluripotency rescue colony scoring was performed on the cell colonies right before alkaline phosphatase assays. Dox treatment of Zhbtc4 cells (inducible Oct4 knockout) results in an almost complete loss of colonies, while dox treatment of 2TS22C cells (inducible Sox2 knockout) results in a high number of flat colonies. Therefore, in Zhbtc4-derived cell lines, pluripotency rescue was scored by measuring the ratio of colonies obtained with and without dox treatment. In 2TS22C-derived cell lines, pluripotency rescue was scored by quantifying the ratio of dome-shaped versus flat colonies.

## DNA constructs and cloning

To generate the pKI Sox2-Nluc-loxP-P2A-Puro-sfGFP-loxP, two multiple cloning sites, the downstream one including a loxP site, were inserted into a pCMV backbone by Oligo annealing. Next, a P2A construct was inserted by oligo annealing between a ClaI and a BamHI site, and the Nluc or Fluc coding sequence fused to a loxP site was cloned between SpeI and ClaI of the resulting plasmid. The selection cassette consisting of sfGFP-Puro or eGFP-Bsd was created using fusion PCR (Hobert, 2002) and inserted between a BamHI and a XhoI site. Subsequently, the Sox2 homology arms (HAs) were inserted. For the 5′ HA, the PCR-amplified 5′ HA and the backbone were digested with BsmBI and the 3′ HA was inserted by In-Fusion cloning into an XbaI site. The pKI Sox1-P2A-Fluc-loxP-pGK-Hygro-loxP construct was based on a previously published plasmid pKI Sox1-P2A-loxP-eGFP-pGK-Hygro-loxP (Deluz et al, 2016), in which eGFP was replaced by Fluc using restriction cloning with AclI and SacI. For all pX330 (Addgene Cat#42230) and pX335 (Addgene Cat#42335) constructs, the vector was opened using BbsI and the guide RNAs were inserted using oligo annealing. The pKI Oct4-HALO-IRES-Bsd was based on pKI Oct4-Fluc-P2A-eGFP-Bsd, in which the Fluc-P2A-eGFP-Bsd cassette was replaced by Halo-IRESbsd using SpeI and NotI. To generate the pKI Sox2-SNAP-IRES-Hygro construct, pKI Sox2-Nluc-loxP-P2A-Puro-sfGFP-loxP was digested with NdeI and re-ligated, thus removing the Nluc-loxP-P2A-Puro-sfGFP cassette and resulting in pKI Sox2-NdeI. pLV pGK-rtTA3G-IRES-Hygro was digested with EcoRI and AgeI to remove the rtTA3G sequence, which was replaced with a P2A sequence by oligo annealing. The resulting vector pLV pGK-P2A-IRES-Hygro9 was digested with XbaI and the coding sequence for the SNAP-tag was inserted, resulting in the pLV pGK-P2A-SNAP-IRES-Hygro construct. Next, the P2A-SNAP-IRES-Hygro cassette was amplified by PCR and used for In-Fusion cloning with pKI-Sox2-NdeI linearized by PCR from the start of the 3′ HA to the end of the 5′ HA of SOX2 (excluding the stop codon). The resulting pKI-Sox2-P2A-SNAP-IRES-Hygro vector was amplified by inverse PCR to remove the P2A-SNAP cassette, which was then replaced by a SNAP-tag using In-Fusion cloning, resulting in the pKI SOX2-SNAP-IRES-Hygro construct.

## Confirmation PCRs

For all knock-in verification PCRs, genomic DNA was purified using a genomic DNA purification kit (Sigma, Cat#G1N350-1KT) and subsequently used to identify clones with correctly targeted alleles. PCR was done using Phusion High Fidelity DNA Polymerase (ThermoScientific, Cat#F530L). Primers are listed in Table 2.

## Western blotting

Cells were trypsinized, collected by centrifugation and washed once in ice-cold PBS. 10 million cells were then resuspended in 500 μl Hypotonic Buffer (20 mM Tris–HCl pH7.4, 10 mM NaCl, 3 mM MgCl$_2$) and supplemented with 1 mM PMSF (AppliChem, Cat#A0999.0005) and Protease Inhibitors (Sigma, Cat#P8340-5ML). After 15 min of incubation on ice, cells were lysed by the addition of 25 μl of 10% NP-40 (Roche, Cat#11.754599001) and subsequent vortexing for 15 s. Nuclei were collected by centrifugation (10 min, 4°C, 16,000 g) and lysed by resuspension in 30 μl RIPA Buffer per million of cells (50 mM Tris–HCl pH7.4, 1% NP-40, 0.5% Sodium Deoxycholate, 0.1% SDS, 2 mM EDTA, 150 mM NaCl), supplemented with PMSF (AppliChem, Cat#A0999.0005) and Protease Inhibitors (Sigma Cat#P8340-5ML). Lysis was allowed to proceed for 30 min during which cells were incubated on ice. Every 10 min, the sample was vortexed and further incubated on ice. To separate soluble nuclear proteins from debris, lysed nuclei were centrifuged for 30 min (4°C, 14,000 rpm). The protein concentration of the supernatant was determined by performing a bicinchoninic acid assay (BCA) (Thermo Fisher, Cat#23235), and 15 μg of protein was mixed with Laemmli sample buffer (Invitrogen, Cat#NP0007) and loaded on an SDS gel (Bio-Rad, Cat#456-1094) for separation (SDS Running Buffer 25 mM Tris, 190 mM Glycine, 0.1%SDS). Proteins were subsequently transferred (Transfer Buffer 25 mM Tris, 190 mM Glycine, 20% Methanol, 0.1% SDS) from the gel onto a PVDF membrane (Merck, Cat#IPVH07850) using a wet transfer system. The membrane was blocked with 5% milk (Roth, Cat#T145.3) in PBS-T to reduce unspecific binding and incubated with the appropriate concentration of primary antibody overnight. The next day, the membrane was rinsed once with PBS-T, incubated with a secondary antibody in 5% milk in PBS-T and washed extensively with PBS-T. Finally, chemiluminescence was revealed using Clarity Western ECL Substrate (Bio-Rad, Cat#170-5060) and the signal was detected on a Fusion FX 7 apparatus (Vilber). The antibodies and concentrations used are summarized in Table 3.

## Quantitative PCR (RT–qPCR)

Total RNA was extracted using the GenElute™ Mammalian Total RNA Miniprep Kit Q-PCR (Sigma, Cat#RTN350), and reverse transcription was performed using an oligoDT primer using superscript II (Thermo Fisher, Cat#18064014). qPCR was performed on a 7900HT Fast Real-Time PCR System (Thermo Fisher) with SYBR green reagent (Roche, Cat#04707516001). The *Rps9* cDNA was used for data normalization. Primers used for RT–qPCR are listed in the Table 4.

## ChIP-seq

The 2TS22C EF1α-YPet-Sox2, Zhbtc4 EF1α-YPet-Oct4 and Zhbtc4 EF1α-Oct4-HALO cells were treated with 1 μg/ml doxycycline for 40–48 h before fixation to ensure complete depletion of endogenous SOX2 (Masui et al, 2007) and OCT4 (Niwa et al, 2000). At least $10^7$ cells were fixed in 1% formaldehyde for 10 min at room temperature, subsequently quenched with 200 mM Tris–HCl pH 8.0, washed with PBS, spun down and stored at −80°C. The cell pellet was then resuspended in 1.5 ml LB1 (50 mM HEPES-KOH pH 7.4, 140 mM NaCl, 1 mM EDTA, 0.5 mM EGTA, 10% Glycerol, 0.5% NP-40, 0.25% Triton X-100), incubated 10 min at 4°C, spun down, resuspended in 1.5 ml LB2 (10 mM Tris–HCl pH 8.0, 200 mM NaCl,

1 mM EDTA, 0.5 mM EGTA) and incubated 10 min at 4°C. The pellet was spun down and rinsed twice with SDS shearing buffer (10 mM Tris–HCl pH 8.0, 1 mM EDTA, 0.15% SDS), and resuspended in 0.9 ml SDS shearing buffer. All buffers contained Protease Inhibitor Cocktail in DMSO (Sigma, Cat#P8340) diluted at 1:100. The suspension was transferred to a milliTUBE 1 ml AFA fibre and sonicated on a E220 focused ultrasonicator (Covaris) using the following settings: 20 min, 200 cycles, 5% duty and 140W. Sonicated chromatin was incubated with 5 µg/$10^7$ cells of the αSOX2 Y-17 (Santa Cruz, Cat#sc-17320) or 5 µl of antibody/4 × $10^6$ cells of the αOCT4A C30A3C1 (Cell Signaling Technology, Cat#5677S) at 4°C overnight. 0.5% BSA-blocked Protein G Dynabeads (Thermo Fischer, Cat#10003D) were added to the chromatin and incubated for 3 h at 4°C. The chromatin was washed several times at 4°C with 5-min incubation between each wash and 2-min magnetization to collect beads; twice with Low Salt Wash Buffer (10 mM Tris–HCl pH 8.0, 150 mM NaCl, 1 mM EDTA, 1% Triton X-100, 0.15% SDS, 1 mM PMSF), once with High Salt Wash Buffer (10 mM Tris–HCl pH 8.0, 500 mM NaCl, 1 mM EDTA, 1% Triton X-100, 0.15% SDS, 1 mM PMSF), once with LiCl Wash Buffer (10 mM Tris–HCl pH 8.0, 1 mM EDTA, 0.5 mM EGTA, 250 mM LiCl, 1% NP-40, 1% sodium deoxycholate, 1 mM PMSF) and with TE buffer (10 mM Tris–HCl pH 8.0, 1 mM EDTA). Beads were subsequently resuspended in Elution buffer (TE buffer with 1% SDS and 150 mM NaCl), treated with 400 ng/ml Proteinase K and reverse crosslinked at 65°C overnight. Samples were purified using MinElute PCR Purification Kit (Qiagen, Cat#28006). For SBR, Zhbtc4 EF1α-YPet-Oct4 and Zhbtc4 EF1α-Oct4-HALO, cells were first fixed with 2 mM Disuccinimidyl glutarate (DSG) (Thermo Fisher, Cat#20593) for 50 min in PBS at room temperature before proceeding with 1% formaldehyde fixation and chromatin immunoprecipitation as described above. One replicate of each ChIP-seq library was prepared with NEBNext ChIP-seq Library Prep Master Mix Set (NEB, Cat#E6240S) using insert size selection of 250 bp. All libraries were sequenced with 75-nucleotide read length paired-end sequencing on a Illumina NextSeq 500 with 30–50 million reads being sequenced for each sample.

### Co-immunoprecipitation

The SBROS and SBR cell lines were plated at a density of about 2 million cells per 10 cm dish, and 48 h later these were treated with 1% formaldehyde for 10 min. Cells were subsequently washed with PBS and then collected in 1 ml lysis buffer (50 mM Tris pH 7.4, 150 mM NaCl, 1% Triton X-100, inhibitors for proteinase and phosphatase). The lysates were then incubated at 4°C with or without αSOX2 antibody (Santa Cruz, Cat#sc-17320) using 5 µg/$10^7$ cells. The next day, 5% BSA-blocked Protein G Dynabeads (Thermo Fischer, Cat#10003D) were added to the samples, which were then washed four times in 1 ml wash buffer (50 mM Tris pH 7.4, 500 mM NaCl, 1% Triton X-100, inhibitors for proteinase and phosphatase). Samples were subsequently incubated at 98°C for 10 min in 40 µl of 2× Laemmli buffer, centrifuged at 2,000 ×g for 3 min and frozen at −80°C. Samples were then used for Western blotting using αOCT4 C-10 antibody at a dilution of 1:200 (Santa Cruz, Cat#sc-5279).

### Immunofluorescence and image acquisition

ES cells were fixed for 15–30 min with ice-cold 2% PFA (Appli-Chem, Cat#A08770500) in PBS, permeabilized and blocked with chilled PBS-Triton (0.5%, AppliChem, Cat#A13880500) and 1% FBS for 30-60 min. Samples were incubated with the primary antibody in PBS and 1% FBS overnight at 4°C, washed twice in PBS and incubated with the secondary antibody in PBS and 1% FBS for 45–60 min. Samples were then washed three times with 0.1% PBS-Tween (Fisher Scientific, Cat#BP337-500), incubated with 1 µg/ml DAPI for 15 min and washed twice with 0.1% PBS-Tween and once with PBS. The antibody dilutions are listed in Table 5.

Immunofluorescence stainings were imaged using a 20× magnification objective (Olympus UPlanSApo 20×, NA 0.75) on an Olympus Cell XCellence or using a 20× magnification objective (Nikon PlanApo 20×, NA 0.75, CFI/60) on a GE InCell Analyzer 2200 apparatus (GE, Cat#29027886, Biomolecular Screening Facility at EPFL).

### Luminescence time-lapse microscopy

Time-lapse luminescence recordings were performed on an Olympus LuminoView LV200 microscope equipped with an EM-CCD cooled camera (Hamamatsu photonics, EM-CCD C9100-13) and a 60× magnification objective (Olympus UPlanSApo 60×, NA 1.35, oil immersion) in controlled environment conditions (37°C, 5% $CO_2$). One day before the experiment, cells were seeded on fluorodishes (WPI, Cat#FD35-100) coated with E-Cadherin in 2 ml of N2B27+2iLIF as described above. For quantitative NANOLUC imaging, knock-in cells were mixed with Calibration cells at a ratio of 1:10 as described previously (Mandic et al, 2017). The medium was supplemented with 1 mM Luciferin (NanoLight Technology, Cat#306A) and 0.5 µl of RealTime Glo Cell Viability Assay Substrate (Promega, Cat#G9711). For imaging in pluripotency maintenance conditions, images were acquired every 299 s in the NANOLUC channel and between 59 and 178 s in Firefly Luciferase channel with a cycle time of 8–17 min for up to 75 h. For overexpression experiments of SOX2-SNAP and YPet-SOX2-delDBD, the exposure time in the NANOLUC channel was 58 s and the cycle time 22 min. For imaging in differentiation conditions, images were acquired every 59 s in the NANOLUC channel and every 599 s in the FLUC channel with a cycle time of 15–16 min for up to 70 h.

For time-lapse imaging of SOX2-NLUC/SOX1-P2A-Fluc, we used an arbitrary threshold of 500 FLUC molecules per cell for at least 4 h to classify cells as Sox1-positive.

### Fluorescence time-lapse microscopy

For time-lapse imaging of the induction kinetics of SOX2-SNAP and YPet-SOX2delDBD as well as for OCT4-HALO single-cell imaging, cells were seeded in E-Cadherin-coated wells of an imaging-grade 96-well plate in N2B27+2iLIF as described above. The next day, the medium was supplemented with 12 nM SNAP-SiR647 dye (NEB, Cat#S9102S) for SOX2-SNAP imaging or 50 nM Halo-SiR ligand (gift from Suliana Manley, EPFL) for OCT4-HALO imaging. Cells were imaged using a 20x magnification objective (Olympus UPlanSApo 20×, NA 0.75) on an Olympus Cell XCellence in controlled conditions (37°C and 5% $CO_2$) for 24 h. For the induction experiments, doxycycline was added to a final concentration of 500 ng/ml to induce transgene expression after 1 h. For the OCT4-HALO imaging, since we observed a mild global fluorescence decrease in the cell population, we corrected for the loss of intensity on the population average to obtain single-cell traces (Fig EV3E and F).

### Flow cytometry and fluorescence-activated cell sorting (FACS)

SBROS cells were stained with the HaloTag TMR Ligand (Promega, Cat#G8251) and the SNAP-SiR647 dye (NEB, Cat#S9102S) at a concentration of 100 nM or 12 nM, respectively, for 1 h. Subsequently, cells were incubated in with Hoechst 33342 (Invitrogen, Cat#H3570) at a concentration of 1.62 µM for 15 min. Cells were then trypsinized, washed in PBS and resuspended in PBS/1% FBS for sorting on a BD FACSAria II.

To determine how endogenous SOX2 levels regulate OCT4 expression, we used the following sorting strategy: Cells were gated for G1 based on Hoechst staining and on a narrow window of intermediate OCT4-HALO expression levels (~25% of cells). This window was further subdivided into two windows defined by the highest or lowest ~30% of SOX2-SNAP expression (Fig EV2L). The converse strategy was used to determine how endogenous OCT4 levels regulate SOX2 expression. After FACS, cells were spun down, resuspended in N2B27 medium and seeded in a gelatinated 24-well plate in N2B27 or N2B27+2iLIF. After 7 h, cells were incubated with both HaloTag TMR and SNAP-SiR647 dyes at a concentration of 100 nM or 12 nM, respectively, for 1 h. Thereafter, cells were again stained with Hoechst 33342 for 15 min and subsequently trypsinized and collected by centrifugation for Flow Cytometry Analysis on a BD Fortessa or BD LSR II, followed by analysis using the FlowJo software (Table EV9).

To evaluate the impact of SOX2 or OCT4 levels on differentiation, we gated cells in G1 based on their Hoechst profile and defined three sub-bins of ~25% in SOX2-SNAP of all G1 cells (Fig EV4A) or into ~25% of OCT4-HALO high and low cells (Fig EV4B), respectively.

For the quadruple sorts based on a combination of SOX2/OCT4 high and low cells, we gated in all G1 or S phase cells on four windows corresponding to ~20% of the total cell population each (Fig EV4F).

Cells were washed once in PBS, trypsinized, collected by centrifugation and resuspended in PBS/1% FBS before flow cytometry analysis. All data acquisition was performed on a BD Fortessa, and analysis was performed using the FlowJo software. E14 cells were used as a negative control to gate for Sox1-eGFP and Bra-mCherry.

### In vitro *differentiation*

For live-cell luminescence microscopy, cells were cultured in N2B27+2iLIF for at least two passages before 30,000 cells were seeded on E-Cadherin in fluorodishes (WPI, Cat#FD35-100) and incubated in N2B27+2iLIF overnight. The next day, the medium was changed to N2B27 supplemented with 1 mM luciferin and 0.5 µl of RealTime Glo Cell Viability Assay Substrate, and image acquisition was started.

For differentiation assays after cell sorting, cells were seeded at a density of 60,000 cells/well of a 6-well plate coated with gelatin. Two days later, the medium was exchanged for fresh N2B27 and after 4 days differentiation outcomes were assessed by flow cytometry on a BD Fortessa.

To direct differentiation towards the mesendoderm, cells were seeded at a density of 60,000 cells/well of a gelatin-coated 12-well plate in N2B27 medium supplemented with 3 µM of CHIR99021. Three days later, differentiation outcomes were assessed by flow cytometry on a BD Fortessa. To direct differentiation towards the

neuroectoderm, cells were seeded at a density of 60,000 cells/well of a gelatin-coated 6-well plate or 30,000 cells/well of a gelatin-coated 12-well plate in N2B27 medium supplemented with 1 µM SB-431542 and 25 ng/ml of bFGF. Four days later, differentiation outcomes were assessed by flow cytometry on a BD Fortessa.

### ATAC-seq

ATAC-seq was performed on 50,000 cells for each condition as previously described (Buenrostro *et al*, 2013). Briefly, 50,000 cells were sorted by FACS, pelleted and washed with 1× ice-cold PBS at 800 *g* for 5 min. Cells were gently resuspended in 50 µl of ice-cold ATAC lysis buffer (10 mM Tris–HCl pH 7.4, 10 mM NaCl, 3 mM MgCl$_2$, 0.1% NP-40) and immediately pelleted at 800 *g* for 10 min at 4°C. To transpose open chromatin regions, cells were resuspended in 50 µl of transposition reaction mix containing 0.5 µM of Tn5 transposase (Chen *et al*, 2017) (gift from Prof. Bart Deplancke Lab, EPFL) in TAPS-DMF buffer (10 mM TAPS-NaOH, 5 mM MgCl$_2$, 10% DMF) and incubated at 37°C for 30 min. The transposed DNA was purified using a DNA purification kit (Zymo Research, Cat#D4003) and eluted in 12 µl of water. A 65-µl PCR was setup with 10 µl of transposed DNA, 0.5 µM of forward primer Ad1_noMX, 0.5 µM of multiplexing reverse primer Ad2.x (Buenrostro *et al*, 2013), 0.6× SYBR® Green I and 1× PCR Master Mix (NEB, Cat#M0544). The samples were thermocycled at 72°C for 5 min, 98°C for 30 s, followed by five cycles of 98°C for 10 s, 63°C for 30 s and 72°C for 1 min. A 15 µl aliquot was analysed by qPCR to determine the number of additional cycles needed to avoid amplification saturation as described in (Buenrostro *et al*, 2013). The amplified ATAC libraries were purified using a DNA purification kit (Zymo Research, Cat#D4003) and size selected using Agencourt AMPure beads (Beckman Coulter, Cat#63881) (0.55× unbound fraction followed by 1.2× bound fraction). All libraries were sequenced with 75-nucleotide read length paired-end sequencing on an Illumina NextSeq 500 with 30–60 million reads being sequenced for each sample.

### Immunofluorescence image analysis

Immunofluorescence images were first background-corrected using the built-in function in the Fiji software. Semi-automated image analysis was then performed using a custom CellProfiler (Kamentsky *et al*, 2011) pipeline. Images were segmented based on their DAPI signal and manually corrected for misidentified objects. Subsequently, fluorescence intensity was measured in the identified nuclei in all channels. The intensities were used to generate histograms of protein expression (NANOG, OCT4 and SOX2), to evaluate the effects of overexpression of OCT4 or SOX2 and to estimate the correlation between OCT4-HALO and total OCT4.

### Cell tracking and single-cell analysis

Cells were tracked manually using Fiji (ImageJ) by defining regions of interest (ROIs) throughout the movie. Next, all ROIs for a single cell were measured and the background (part of the image in the vicinity of the cell but devoid of cells) was subtracted. We used a previously reported method to convert the observed light intensity to absolute molecule numbers (Mandic *et al*, 2017).

To determine SOX2 levels in pluripotency conditions, cells were *in silico* synchronized for cell cycle progression using linear interpolation of the time variable, and absolute molecule numbers were

converted to nuclear concentration, using a model for the nuclear size increase during the cell cycle (Filipczyk *et al*, 2015) and a reported estimate of the nuclear volume of ES cells (Chalut *et al*, 2012). To evaluate how cells readjust their SOX2 levels over time, we used a rank-based autocorrelation (Sigal *et al*, 2006) using data from cells tracked over one or two consecutive cell cycles. To compare the autocorrelation function between data tracked for one and two cell cycles, we selected 100 random single-cell traces from the SOX2 data and calculated the protein memory based on a conservative mixing time estimation (Sigal *et al*, 2006). As the results using data from one and two cell cycles were similar, we used a single-cell cycle from the OCT4-HALO imaging to calculate the rank-based autocorrelation of OCT4-HALO.

To determine how SOX2 levels predict neuroectodermal differentiation, we classified tracked cell cycles based on their FLUC signal in four groups, using an arbitrary threshold of 500 AU in FLUC maintained for at least 4 h: "negative cells" were defined as cells below the threshold throughout the movie; "before SOX1$^+$" were defined as negative cells that pass the threshold in the next cell cycle; "turning SOX1$^+$" were defined as cells passing the threshold in the current cell cycle; "SOX1$^+$" were defined as cells with FLUC levels above the threshold. The "before SOX1$^+$" cell population also contains traces that did not cover a full cell cycle before becoming Sox1-positive. All single-cell traces were *in silico* synchronized using a linear interpolation of the time variable. A two-sided *t*-test with unequal variance was performed for the mean SOX2 levels in the cell cycle before cells turn SOX1 positive to evaluate statistical significance (Table EV10).

To determine the induction kinetics in the YPet-SOX2-delDBD- and SOX2-SNAP-overexpressing cell lines, single cells were tracked over divisions in one daughter cell.

### ATAC-seq and ChIP-seq analysis

ATAC-seq and ChIP-seq reads were aligned to the mouse reference genome mm10 using STAR 2.5.3a (Dobin *et al*, 2013) with settings "–alignMatesGapMax 2000 –alignIntronMax 1 –alignEndsType EndtoEnd –outFilterMultimapNmax 1". Duplicate reads were removed with Picard (Broad Institute), and reads not mapping to chromosomes 1–19, X or Y were removed. For each sample, peaks were called with MACS 2.1.1.20160309 (Zhang *et al*, 2008) with settings "-f BAMPE -q 0.01 -g mm". For ATAC-seq data, peaks from all samples were merged with BEDOPS (Neph *et al*, 2012). Peaks overlapping peaks called for ChIP-seq Input data from asynchronous mouse ES cells (GSE89599) were discarded. The HOMER2 (Heinz *et al*, 2010) function annotatePeaks.pl was used with settings "-noadj -len 0 -size given" to count the number of reads for each sample in peaks. TMM normalization was done with edgeR (Robinson *et al*, 2010) and analysis of differentially abundant regions with limma (Ritchie *et al*, 2015). The analysis was done using three different contrasts: (i) SHOH versus SLOL, design ~0+Condition+Replicate, (ii) SOX2 high versus SOX2 low, design ~0+SOX2+OCT4+Replicate, (iii) OCT4 high versus OCT4 low, design ~0+OCT4+SOX2+Replicate. Regions with an adjusted *P*-value < 0.1 for at least one test were used in the analysis. Groupings were made according to fold-change direction and if loci were significantly different for OCT4 high versus OCT4 low only (OCT4 regulated), SOX2-high versus SOX2-low only (SOX2 regulated), or regulated by both SOX2 and OCT4 or OCT4-high/SOX2-high versus OCT4-low/SOX2-low (co-regulated). SOX2

and OCT4 peaks used to determine overlap were taken from GSE87822 and GSE92846 (King & Klose, 2017; Liu *et al*, 2017b). Additional OCT4 ChIP-seq data in Fig EV5K were taken from GSE78073 (Shin *et al*, 2016) and GSE56138 (Buecker *et al*, 2014). H3K4me3 peaks in ES-Bruce4 cells from ENCODE (ENCODE Project Consortium, 2012) and ES cell super-enhancers from (Whyte *et al*, 2013) were converted to mm10 using liftOver (Hinrichs *et al*, 2006). Gene ontology analysis was done using the closest UCSC-annotated gene to each peak with Fisher's exact test in topGO using genes closest to all peaks as background. bigWig files for both ATAC-seq and ChIP-seq were generated by merging replicate bam files with SAMTools (Li *et al*, 2009) followed by the deepTools 2.4.2 (Ramírez *et al*, 2014) functions bamCoverage (with setting "–normalizeUsingRPKM"). Average lineplots and ChIP-seq heatmaps were generated using deepTools computeMatrix (with setting "reference-point") and plotHeatmap. The WT SOX2 ChIP-seq profile in Fig EV1I is based on data from GSE89599 (Deluz *et al*, 2016). Genome tracks were generated in the UCSC genome browser (Kent *et al*, 2002). Motif enrichment was calculated using the HOMER2 function find-Motifs.pl with setting "-size given". The most frequent known motif corresponding to OCT (OCT4) and SOX (SOX3) family motifs was used. Background enrichment was calculated as the mean of the HOMER-estimated background frequency in all groups. For the random forest model (Fig EV5M and Table EV14), we first batch downloaded all peak files from cistromeDB (http://cistrome.org/db) (Mei *et al*, 2017) in the Mouse_Factor and Mouse_Histone categories. For the groups in Fig 4B, we annotated genomic regions overlapping any of the regions in the peak files from cistromeDB and kept only those where Cell_type was "Embryonic Stem Cell". We generated a data frame in R with the group belonging of each region and the overlap status (1/0) with the different peak files, consisting 3,725 columns (one for group belonging and 3,724 for the peak identities) and 81,132 rows (one for each region). We used a subset so that each group was represented by the same number of regions, i.e. the lowest number of regions in a group (n = 209). These data were split into training (80%, n = 1,170) and test (20%, n = 293) regions and the training regions were used to train a random forest model. The function randomForest was used with the settings "formula = Group~., mtry = 4, ntree = 2001", where Group is the column containing the group identity of the region. The model was used on the testing data using the predict function with setting "type = 'response'".

### Determination of SOX2 and OCT4 protein half-lives

SBROS cells were seeded at 30,000 cells/cm$^2$ on E-cadherin as described above (for SBROS- and SBROS-overexpressing SOX2-SNAP) or on StemAdhere (Primorigen, Cat#S2071-500UG; for SBROS-overexpressing YPet-SOX2-delDBD) in N2B27+2iLIF. Briefly, StemAdhere diluted 1:25 in PBS (with Ca$^{2+}$ and Mg$^{2+}$; Amimed, Cat#3-05K00-I) was added to the culture vessel and incubated for 60 min at 37°C. Just before seeding, the StemAdhere solution was removed, and the surface of the vessel rinsed once with PBS and filled with the appropriate cell culture medium. After 24 h of cell culture, cells were pulse-labelled with 12 nM of SNAP-SiR 647 or different concentrations of Halo-SiR ligand (see below) (gift from Suliana Manley, EPFL) for 30 min at 37°C. In addition, for OCT4-HALO half-life determination as a function of SOX2-SNAP levels, cells were initially stained with the SNAP-Cell® TMR-Star dye (NEB) at a final concentration of 3 μM and imaged in

the Cy3 channel (excitation filter 542/27 nm, emission filter 597/45 nm). For OCT4-HALO half-life determination upon YPet-SOX2-delDBD overexpression, the YPet channel (excitation filter 513/17 nm, emission filter 548/22 nm) was initially imaged and only YPet-positive cells were analysed for Halo-SiR signal decay. Cells were then washed 3× with PBS and incubated in N2B27+2iLIF medium for 15 min at 37°C. This washing step was repeated once more, and then, cells were washed 2× with PBS and phenol-free N2B27 medium+2iLIF was added. Note that we observed variations in the signal strength of the Halo-SiR that may arise from batch-to-batch differences or loss of signal due to storage. To account for this, we tested several dye concentrations and used the ones that yielded clear exponential decays (too much dye cannot be washed out and the signal does not decay (Alber & Suter, 2018)). These concentrations varied between 0.2 nM and 20 nM Halo-SiR. The decay of the fluorescence signal was imaged using an InCell Analyzer 2200 microscope (GE Healthcare Life Sciences) with a 20× Objective, 10% laser power, 300-ms exposure (Cy5: excitation filter 632/22 nm, emission filter 679/34 nm) and 2 × 2 binning for 12.5 h at intervals of 15 min. Images were analysed in FiJi, where the background was subtracted from all images in the stack (rolling ball radius = 50 pixels). The integrated fluorescence intensity was then quantified by manual tracking. ROIs were drawn around each cell of interest at each time point, and the integrated fluorescence intensity was calculated by multiplying the area of each ROI with its mean intensity. The local background was calculated by drawing a ROI close to the cell of interest in each time frame, followed by multiplying its mean intensity with the area of the cellular ROI. The background intensity was subtracted from the cellular intensity for each time frame. In case of cell divisions, both daughter cells were tracked separately and their integrated intensities were summed. Each fluorescence intensity trace was normalized to the value of the first frame and the single-cell decay rates (b) were determined by exponential curve fitting, using the curve fitting tool in Matlab (Fig 1E) or the nls function in R (Fig 2C) (fitted equation: $f(t) = e^{-bt}$). Half-lives were then calculated as follows: $t_{1/2} = b/\ln(2)$. Single-cell half-lives in 20 cells were quantified for SOX2-SNAP, OCT4-HALO and OCT4-HALO after 6–8 h of SOX2-SNAP or YPet-SOX2-delDBD overexpression.

### Statistical analysis

Statistical testing was done with either *t*-tests with unequal variance (Welch) or, when a Shapiro–Wilk's test indicated non-normality ($P < 0.05$), Mann-Whitney *U*-tests, as described in the Figure legends. For Fig 3C, two-sided *t*-tests with unequal variance were performed for each cell cycle progression (time) point, see Table EV10 for *P*-values. For the autocorrelation functions in Figs 2I and L, and EV3D, the error bars denote the SE estimated by bootstrapping. For Fig EV2K, the *P*-value is based on Pearson correlation.

## Data availability

ATAC-seq and ChIP-seq data: GEO (Gene Expression Omnibus) GSE126554 (https://www.ncbi.nlm.nih.gov/geo/query/acc.cgi?acc=GSE126554).

   The semi-automated image analysis pipeline is provided as Computer Code EV1.

**Expanded View** for this article is available online.

## Acknowledgements

This work was supported by the Swiss National Science Foundation (PP00P3_1144828 and PP00P3_172905) to D.M.S, the Pierre Mercier Foundation and the generous support of the Fondazione Teofil Rossi di Montelera e di Premuda and an anonymous donor advised by CARIGEST SA. We thank the Swiss Federal Institute of Technology (EPFL), the Biomolecular Screening Facility (EPFL-BSF) and the EPFL Bioimaging and Optics Core Facility (EPFL-BIOP) for assistance in imaging and the EPFL Flow Cytometry Core Facility (EPFL-FCCF) for the fluorescence-activated cell sorting. We thank Antonio Meireles-Filho for help with ATAC-seq experiments.

## Author contributions

Conceptualization, DS and DMS; Methodology, DS, DMS, ETF, SG, ABA; Formal Analysis, DS, DMS, ETF, ABA; Investigation, DS, DMS, CD, ETF, SG, ABA; Resources, DMS; Writing – Original Draft, DS and DMS; Writing – Review & Editing, DS, DMS, ETF, CD, SG, ABA, Funding Acquisition, DMS; Supervision, DMS.

## Conflict of interest

The authors declare that they have no conflict of interest.

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
