## [Review Process File · Molecular Systems Biology]

Endogenous fluctuations of OCT4 and SOX2 bias pluripotent cell fate decisions

Daniel Strebinger, Cédric Deluz, Elias T. Friman, Subashika Govindan, Andrea B. Alber and David M. Suter.

Review timeline:

Submission date:	13 th May 2019
Editorial Decision:	27 th June 2019
Revision received:	9 th August 2019
Editorial Decision:	30 th August 2019
Revision received:	2 nd September 2019
Accepted:	4 th September 2019

Editor: Maria Polychronidou

Transaction Report:

1st Editorial Decision

27th June 2019

Thank you again for submitting your work to Molecular Systems Biology. We have now received the reports of two of the three reviewers who were asked to evaluate your manuscript. In the interest of time, we have decided to proceed with making a decision based on these two reports. As you will see below, the reviewers acknowledge that the presented findings seem potentially interesting. They raise however a series of concerns, which we would ask you to address in a major revision.

Without repeating all the points listed below, one of the more fundamental issues raised refers to the need to include follow up analyses and validations in order to strengthen the main conclusions and enhance the impact of the study. Both reviewers provide constructive suggestions in this regard.

All other issues raised by the reviewers need to be satisfactorily addressed. As you may already know, our editorial policy allows in principle a single round of major revision so it is essential to provide responses to the reviewers' comments that are as complete as possible. Please feel free to contact me in case you would like to discuss in further detail any of the issues raised by the reviewers.

REFEREE REPORTS

Reviewer #1:

Summary

In this manuscript, Strebinger et al. assessed the endogenous fluctuations of the pluripotency factors OCT4 and SOX2 and their effect on the differentiation capacity of mouse embryonic stem cells (ESCs). To do so, they created various overexpression constructs and knock-in reporter cell lines of

mouse ESCs, and used these lines to quantify and track protein levels in single cells and in the course of 2 cell divisions. Importantly, using these reporter lines the authors were able to sort cells expressing high or low levels of OCT4 and SOX2 to test their relative differentiation potential. They also performed ATAC-seq to characterize the effect of these endogenous protein fluctuations on the chromatin landscape. The authors found that SOX2 overexpression negatively affects the transcriptional levels of SOX2 supporting a negative feedback loop, while increases OCT4 protein levels by increasing its half-life. They determined that both proteins fluctuate rapidly, 2-3 fold over the course of one cell cycle, and that their levels during the G1 cell cycle phase introduce lineage differentiation biases (high SOX2 primes for neuroectoderm (NE) while high OCT4 primes for both NE and mesendoderm (ME) fates). These fluctuations were also shown to affect chromatin accessibility, with high OCT4 levels increasing accessibility at enhancers nearby to differentiation-associated genes.

General Remarks

Many of the key conclusions are convincing. The reporter lines are elegant, powerful and thoroughly tested and validated for proper functioning of both SOX2 and OCT4 either as part of this or previous publications from the group. The use of knock-in fusion cell lines allows for quantitative and live tracking of the endogenous levels of both SOX2 and OCT4, a benefit of the study, although the heterozygous nature of the OCT4 fusion is a limitation that must be kept in mind. This study sheds light on the dynamic nature of these two important pluripotency factors, and furthers our understanding of how endogenous fluctuations may impact cell fate decisions. It hints at a possible mechanism by which the levels of these proteins may affect pluripotency (changes in chromatin accessibility), but it does not do any functional validations or explore why this would be relevant in G1 but not S phase of the cell cycle. Most of the author's conclusions are orthogonal validations of previous knowledge (not always cited properly), while few findings that contradict previous work are not discussed or validated further. In general, this is an elegant and mostly descriptive study of interest to stem cell biologists and those interested in the use of imaging and cell tracking to study protein dynamics.

Major Points

- Many of the results are redundant or confusing when it comes to comparisons of the inducible overexpression systems with the sorted high and low expressing cells. The need of using and reporting both approaches is not always clear to me. If the authors want to validate the effects of high OCT4 or SOX2 expression on other pluripotency markers or on lineage bias they could start by presenting the results of sorted cells and then showing that their overexpression system achieves similarly increased protein levels (which is currently missing e.g. in Fig.3-f) and similar (or not) results.
- Fig 3g: The discussion regarding the OCT4 overexpression experiment is somewhat confusing. The authors claim that the OCT4 OE may affect lineage bias differently than the endogenous OCT4 fluctuations due to "opposing roles of high OCT4 levels before and after removal of pluripotency signals," however, in both cases, the cells are sorted in G1 and then plated into no 2i/LIF media, and so the pluripotency signals are removed at the same time.
- The observation that SOX2 OE affects OCT4 protein stability is an interesting and novel -to my knowledge- finding. The authors could elaborate on the mechanism by at least testing if this is dependent on either the DNA binding domain of Sox2 or its well-characterized OCT4 heterodimerization region.
- The tight regulation of OCT4 protein and the drastic effects of upregulation or downregulation of OCT4 in stemness have been previously characterized. Already in 2000, Niwa et al, reported that <2fold elevation of OCT4 levels induced increased mesoderm differentiation, while OCT4 reduction turns on the trophoectoderm program. The authors should properly discuss their findings in the context of previous work and importantly they should include trophoectoderm lineage markers as an expected outcome of OCT4-low expressing cells.
- The authors should expand their ATAC-seq analysis by at least performing motif enrichment analysis around OCT4 regulated, SOX2 regulated and unaffected regions and potentially unbiased enrichment analysis using the wealth of TF and histone mark ChIP-seq datasets in ESCs. This could help to validate their findings and possibly identify binding partners that have a role in the increase or decrease in accessibility.

Minor Points

- The histogram for supplemental figure 2l is slightly confusing. It suggests that the cells are

sorted/gated again for G1 when performing the analysis of OCT4/SOX2 levels, 8 hours after the G1 sorting and replating. By this time, most of the cells should have left G1 and entered S phase, so this focus on G1 would skew the data towards cells that have arrested or are cycling more slowly.

- A few of the gene ontology graphs have lines overlapping that make them challenging to read
- Supp Fig 5j seems to have an incorrect description in the figure legend that doesn't match with what appears in the figure
- Fig 4E: Given these results, it seems relevant for the authors to characterize the percent of accessible regions that are at the TSS, intergenic, and intragenic for each of the 7 categories in this figure.

Reviewer #2:

Strebinger et al. study how variations in the levels of the pluripotency factors Oct4 and Sox2 impact the differentiation of mouse embryonic stem cells to neuroectoderm or mesendoderm. The authors have an interesting set of ES lines in which they can monitor Oct4, Sox2 and two differentiation markers (Sox1 and Brachyury) in single cells or vary pluripotency factor levels in an inducible manner. They find that Sox2 represses its own expression and induces Oct4 expression. As Sox2 and Oct4 form heterodimers, it would be of particular interest to study the correlation between Sox2 and Oct4 fluctuations in single cells and the impact of differential Sox2/Oct4 levels on differentiation (both of which the authors can easily do with their cell line). The authors also suggest that Sox2 and Oct4 levels influence the commitment of ES cells. This part of the study is potentially very interesting but is supported by much weaker and contradicting evidence.

Major concerns:

1. The differentiation experiments show extensive variability in the fraction of Brachyury positive cells between assays. These differences cannot be explained by the sorting schemes themselves. Indeed in Supplementary Fig. 4a, 62 to 78% of cells are Brachyury positive whereas control of other assays have fewer Bra+ cells. It is also worrisome for the reproducibility of the assays that the no Dox controls in Supplementary Fig. 4c and 4d are very different. In order to believe the conclusions, the data should show internal consistency between controls.
2. Halo-TMR emission spectrum overlaps significantly with mCherry's, questioning its use in the SBROS line. The authors should demonstrate that the labeling schemes they use for the SNAP and Halo tags don't interfere with the measurement of mCherry by using other substrates (for example a blue one).
3. The authors find that high Oct4 levels enhance neuroectoderm commitment in contradiction with a previous report. To further substantiate their claim, they should study how Oct4 and Sox2 fluctuations impact directed differentiation to neuroectoderm. It will also add generality to the authors' findings.
4. While Sox2 has a constant production rate during the cell cycle, Oct4 has an interesting production profile (Supplementary Fig. 3). The data in Fig. 2j suggests quite some heterogeneity between single cells. What is the heterogeneity in the production rate of Sox2 and Oct4 during the cell cycle in single cells? How do the production rates of Sox2 and Oct4 correlate in single cells? How does the ratio between Sox2 and Oct4 vary in individual cells during the cell cycle?
5. Does the Oct4/Sox2 ratio influence commitment?

Minor concern:

The readability of Fig. 2 and 3 could be improved by choosing more contrasting colors and adding extra labels.

Reviewer #1:

Summary

In this manuscript, Strebinger et al. assessed the endogenous fluctuations of the pluripotency factors OCT4 and SOX2 and their effect on the differentiation capacity of mouse embryonic stem cells (ESCs). To do so, they created various overexpression constructs and knock-in reporter cell lines of mouse ESCs, and used these lines to quantify and track protein levels in single cells and in the course of 2 cell divisions. Importantly, using these reportets the authors we able to sort cells expressing high or low levels of OCT4 and SOX2 to test their relative differentiation potential. They also performed ATAC-seq to characterize the effect of these endogenous protein fluctuations on the chromatin landscape. The authors found that SOX2 overexpression negatively affects the transcriptional levels of SOX2 supporting a negative feedback loop, while increases OCT4 protein levels by increasing its half-life. They determined that both proteins fluctuate rapidly, 2-3 fold over the course of one cell cycle, and that their levels during the G1 cell cycle phase introduce lineage differentiation biases (high SOX2 primes for neuroectoderm (NE) while high OCT4 primes for both NE and mesendoderm (ME) fates). These fluctuations were also shown to affect chromatin accessibility, with high OCT4 levels increasing accessibility at enhancers nearby to differentiation-associated genes.

General Remarks

Many of the key conclusions are convincing. The reporter lines are elegant, powerful and thoroughly tested and validated for proper functioning of both SOX2 and OCT4 either as part of this or previous publications from the group. The use of knock-in fusion cell lines allows for quantitative and live tracking of the endogenous levels of both SOX2 and OCT4, a benefit of the study, although the heterozygous nature of the OCT4 fusion is a limitation that must be kept in mind. This study sheds light on the dynamic nature of these two important pluripotency factors, and furthers our understanding of how endogenous fluctuations may impact cell fate decisions. It hints at a possible mechanism by which the levels of these proteins may affect pluripotency (changes in chromatin accessibility), but it does not do any functional validations or explore why this would be relevant in G1 but not S phase of the cell cycle.

We thank the reviewer for her/his overall assessment of our work. We also take the opportunity to clarify our findings regarding the following sentence: "It hints at a possible mechanism by which the levels of [OCT4 and SOX2] may affect pluripotency (changes in chromatin accessibility)". In fact, we did not observe differences in chromatin accessibility of pluripotency-associated gene regulatory elements that depend on low/high OCT4 and/or low/high SOX2 levels. However, we did find that chromatin accessibility of differentiation-associated enhancers bound by OCT4 was associated with low/high OCT4 levels.

We indeed did not attempt at a mechanistic understanding of the differential effect of OCT4 and SOX2 concentration variations between G1 and S phase. While this would certainly be of interest, we believe that this is out of the scope of the present study.

Most of the author's conclusions are orthologous validations of previous knowledge (not always cited properly), while few findings that contradict previous work are not discussed or validated further.

We agree that some of our findings validate previous knowledge. We have already cited several previous studies, which we have now discussed more extensively in our revised manuscript (page 7, 12 and 17). However, our study also describes a number of novel findings, and the central one is the discovery that endogenous fluctuations of transcription factor levels within a phenotypically homogeneous ES cell population impact cell fate decisions towards mesendoderm and neuroectoderm. This is to our knowledge unprecedented, and is conceptually very different from previous work on NANOG fluctuations that stem from transitions of ES cells between different phenotypic states. Below we summarize the main new findings reported in our study:

1. Positive regulation of OCT4 protein stability by increased SOX2 protein levels (Fig.2C and revised Fig EV2K)
2. Control of endogenous OCT4 protein levels by endogenous SOX2 protein level fluctuations (Fig 2E)
3. Quantitative analysis of endogenous fluctuations of SOX2 and OCT4 protein levels on the timescale of the cell cycle (Fig 2G-L)
4. Upon undirected differentiation, increased capacity of cells with endogenously high OCT4 levels to differentiate towards the neuroectoderm and the mesendoderm, and of cells with endogenously high SOX2 levels to differentiate towards the neuroectoderm (Fig 3E, F and I)
5. Cell cycle-dependent OCT4/SOX2 level differentiation bias (observed in G1 phase but not in S phase, Fig 3I and J, respectively)
6. Opening of differentiation-associated enhancers in G1 cells associated to high endogenous OCT4 levels (Fig 4)

There are a few places in which our study may appear to contradict previous studies, and here we would like to provide some clarifications:

1. The impact of OCT4 levels on differentiation. We have found that endogenously high OCT4 levels before differentiation results in increased neuroectodermal commitment; this contrasts with OCT4 overexpression or high endogenous OCT4 expression levels throughout differentiation, which results in inhibition of neuroectodermal commitment (Thomson et al., Cell 2011). Importantly, our own experiments of OCT4 overexpression throughout differentiation are in complete agreement with previous reports (Fig EV4E or our manuscript showing similar findings than Thomson et al., Cell 2011). Thus there is no clear contradiction here and our results show that the impact of OCT4 levels is highly context-dependent. We have now clarified this in the discussion section on page 17 of the revised manuscript.
2. A positive regulation of *Sox2* transcription by SOX2 had been suggested because:
 - i) The SOX2/OCT4 heterodimer is known to bind to *Sox2* regulatory sequences. However, the mere fact that SOX2 binds to *Sox2* regulatory regions does not constitute any functional evidence for positive regulation of *Sox2* transcription by SOX2.
 - ii) Prolonged *Sox2* mRNA depletion using RNA interference decreases *Sox2* promoter activity. This approach does not allow to determine the direct impact of SOX2 on *Sox2* mRNA expression since prolonged knockdown of *Sox2* leads to pluripotency exit (Masui et al., Nat Cell Biol 2007). In contrast, our short-term *Sox2* overexpression experiments allowed us to assess the direct impact of SOX2 on *Sox2* mRNA levels, and in fact our findings are in line with two other studies (Kopp et al, 2008; Ormsbee Golden et al, 2013). We have now discussed this more extensively on page 7 of the revised manuscript and cited the appropriate literature.

In general, this is an elegant and mostly descriptive study of interest to stem cell biologists and those interested in the use of imaging and cell tracking to study protein dynamics.

We thank the reviewer for her/his positive assessment of our work.

Major Points

- Many of the results are redundant or confusing when it comes to comparisons of the inducible overexpression systems with the sorted high and low expressing cells. The need of using and reporting both approaches is not always clear to me. If the authors want to validate the effects of high OCT4 or SOX2 expression on other pluripotency markers or on lineage bias they could start by presenting the results of sorted cells and then showing that their overexpression system achieves similarly increased protein levels (which is currently missing eg in Fig.3-f) and similar (or not) results.

We thank the reviewer for this opportunity to provide some clarifications. Both overexpression and analysis of endogenous fluctuations are required as they complement each other for reasons I explain below. Overexpression allows a functional assessment of the impact of SOX2 and OCT4 level changes, but brings their expression levels out of their physiological range. Thus, these experiments alone do not provide direct evidence that the endogenous, physiological variations of OCT4/SOX2 levels occur in a regime in which these impact differentiation outcomes. We thus do not claim that our overexpression system « achieves similarly increased protein levels », and it would not be a realistic goal. In contrast, sorting cells for their endogenously high or low SOX2/OCT4 expression levels followed by analysis of differentiation outcome is a correlative type of experiment, but does reflect physiologically relevant conditions. Thus these two approaches are complementary and when considered together, shed light on the functional impact of endogenous SOX2/OCT4 fluctuations on differentiation.

- Fig 3g: The discussion regarding the OCT4 overexpression experiment is somewhat confusing. The authors claim that that the OCT4 OE may affect lineage bias differently than the endogenous OCT4 fluctuations due to "opposing roles of high OCT4 levels before and after removal of pluripotency signals," however, in both cases, the cells are sorted in G1 and then plated into no 2i/LIF media, and so the pluripotency signals are removed at the same time.

Let us clarify this point. When we overexpress OCT4 exogenously, we reach suprphysiological OCT4 levels since we express additional OCT4 molecules on top of endogenously expressed OCT4. Importantly, when we give a brief pulse of dox before differentiation we subsequently withdraw dox, 2i and LIF at the same time, and thus suprphysiological OCT4 levels will persist for some time after the onset of differentiation. This contrasts with high endogenous OCT4 levels without overexpression, which by definition are within the OCT4 physiological range in ES cells. It is unfortunately very challenging to fully mimic OCT4 fluctuations using a controllable system, and thus OCT4 overexpression suffers from this limitation. Nevertheless, we found that a brief pulse of OCT4 overexpression before differentiation:

- i) increases ME differentiation (Fig 3G) similarly to ES cells sorted for high OCT4 levels (Fig 3F and I) or overexpressing OCT4 throughout differentiation (Fig EV4E).
- ii) has an intermediate effect on NE differentiation (Fig 3G) as compared to cells sorted for high OCT4 levels (increased NE differentiation, (Fig 3F and I) vs OCT4 overexpression throughout differentiation (decreased NE differentiation, (Fig EV4E)).

Thus our results show that high endogenous OCT4 levels at the onset of differentiation enhance neuroectodermal differentiation, but that high OCT4 levels during differentiation inhibit neuroectodermal differentiation (as also shown by Thomson et al., Cell 2011). We have now edited the manuscript to further clarify this point (see page 12 of the revised manuscript).

- The observation that SOX2 OE affects OCT4 protein stability is an interesting and novel -to my knowledge- finding. The authors could elaborate on the mechanism by at least testing if this is dependent on either the DNA binding domain of Sox2 or its well-characterized OCT4 heterodimerization region.

This is an interesting point. To satisfy the reviewer's request, we have now generated a SBROS cell line allowing inducible overexpression of SOX2 lacking its DNA-binding domain (YPet-SOX2delDBD). We found that in contrast to WT SOX2 overexpression, SOX2delDBD did not stabilize OCT4 but instead led to a minor decrease in OCT4 half-life (revised Fig 2C). We now also show that SOX2-SNAP expression levels are positively correlated to OCT4-HALO half-life at the single cell level (revised Fig EV2K).

- The tight regulation of OCT4 protein and the drastic effects of upregulation or downregulation of OCT4 in stemness have been previously characterized. Already at 200, Niwa et al, reported that <2fold elevation of OCT4 levels induced increased medoderm differentiation, while OCT4 reduction turns on the trophoectoderm program. The authors should properly discuss their findings in the context of previous work and importantly they should include trophoectoderm lineage markers as an expected outcome of OCT4-low expressing cells.

We thank the reviewer for the opportunity to clarify this point. Indeed, elevation of OCT4 levels were already shown before to increase mesoderm differentiation, however this was in the context of OCT4 overexpression throughout differentiation. Note that we have also found that constant OCT4 overexpression during differentiation increases mesendodermal commitment (Fig EV4E). However, the novel aspect in our study is the finding that physiological fluctuations of OCT4 levels before differentiation onset are sufficient to mediate these effects. Concerning the inclusion of trophoctoderm lineage markers, we would also like to provide some clarifications. Niwa et al. indeed reported increased trophoctoderm differentiation in the context of complete suppression of OCT4 expression. In contrast, ES cells heterozygous for OCT4, which show reduced OCT4 levels (about 65% of the average wt levels, Niwa et al., Nat. Genetics 2000) display increased maintenance of the pluripotent state and do not show any evidence of increased trophoctodermal differentiation (Karwacki-Neisius et al., Cell Stem Cell 2013). Since OCT4-low and OCT4-high cell populations differ by less than 2-fold in their OCT4 levels (Fig EV4B), i.e. corresponding to > 66% and < 133% of the average OCT4 levels in the whole population, respectively, we do not expect OCT4-low cells to turn on trophoctoderm lineage markers.

We also compared the OCT4 expression level of ZHBTc4 cells used in the study by Niwa et al., and SBROS cells by quantitative immunofluorescence without dox or with dox treatment for different time durations (Figure 1 here below). We found that in the absence of doxycycline, ZHBTc4 expressed on average 36% of the average OCT4 expression levels in SBROS cells. Therefore: i) OCT4 levels in untreated ZHBTc4 cells are substantially lower than our OCT4-low cells from the SBROS cell line, and ii) OCT4 levels in ZHBTc4 cells are permanently lower, in contrast to OCT4-low cells from the SBROS cell line that is a temporary state, as illustrated by the fluctuation timescales of OCT4 levels shown in Fig 2K and L of our manuscript. According to Niwa et al., Nature Genetics 2000, Fig.4, the ZHBTc4 cell line does not turn on trophoctodermal lineage markers in the absence of doxycycline. Thus, even 36% of WT OCT4 levels are sufficient to robustly maintain the pluripotent state.

Thus, we believe that the expression of trophoctodermal lineage markers is in fact not “an expected outcome of OCT4-low expressing cells”. We have now discussed the relationship between OCT4 levels and trophoctoderm, pluripotency and neuroectodermal/mesendodermal differentiation more in detail on page 17 of the revised manuscript.

Figure 1: OCT4 levels as measured by immunofluorescence in ZHBTc4 cells that were untreated (null), or treated with 1 μ g/ml of dox for four, six or eight hours, compared to OCT4 levels in the SBROS cell line. Number of cells analysed: null: n=199 ; four: n=254 ; six: n=586 ; eight: n=595 ; SBROS: n=310.

- The authors should expand their ATAC-seq analysis by at least performing motif enrichment analysis around OCT4 regulated, SOX2 regulated and unaffected regions and potentially unbiased enrichment analysis using the wealth of TF and histone mark ChIP-seq datasets in ESCs. This could help to validate their findings and possibly identify binding partners that have a role in the increase or decrease in accessibility.

We thank the reviewer for this suggestion, and we have now performed the requested analysis. As expected, we found the OCT motif to be most enriched in the group losing accessibility upon OCT4 loss, and vice versa for the SOX motif and SOX2 loss (revised Fig EV5L). As suggested, we have taken advantage of the large number of available sequencing datasets in ES cells. We downloaded peak files from the database cistromeDB, which applies a uniform pipeline to analyse publicly available data, and extracted 3'724 peak datasets from ES cells, excluding ATAC-seq and DNase-seq data. We then checked for overlap in our identified groups as well as in unaffected accessible regions and trained a model using random forest that could relatively well separate upregulated and downregulated regions affected by OCT4 and SOX2 levels (revised Fig EV5M). While as expected, regions with decreased accessibility in OCT4 and SOX2 high cells (downregulated) were enriched for promoter marks, those with increased accessibility (upregulated) were enriched for overlap with NANOG. Inspection of the NANOG binding profile in these regions confirmed the enrichment. Thus, we believe that we have identified a possible binding partner at these regions. NANOG has recently been shown to influence the chromatin accessibility landscape in ES cells at OCT4 and SOX2 sites (Heurtier et al. Nature Communications, 2019), in line with this idea. We have included this data as Fig EV5M and Appendix Table S13, including descriptions in the main text (page 15-16 of the revised manuscript) and Methods (page 35).

Minor Points

- The histogram for supplemental figure 2l is slightly confusing. It suggests that the cells are sorted/gated again for G1 when performing the analysis of OCT4/SOX2 levels, 8 hours after the G1 sorting and replating. By this time, most of the cells should have left G1 and entered S phase, so this focus on G1 would skew the data towards cells that have arrested or are cycling more slowly.

We thank the reviewer for the comment, and we apologize for this point that was indeed a bit confusing. We gate on cells in G1 phase even after the eight-hour incubation to ensure that we are not biasing the quantitative analysis of OCT4 and SOX2 expression due to cell cycle-related differences in cell size. Importantly, after sorting ES cells typically remain at least 4-6 hours in suspension before they re-attach, and this delays their entry into S-phase. Therefore, in these conditions a large fraction of the cells have actually not left G1 as can be seen on the flow cytometry diagram of Fig EV2M. Even though there are cells that progressed beyond G1 in this time frame and thus our analysis moderately skews the data towards slowly cycling cells, this does not affect our conclusions. We have now clarified this in the revised manuscript (page 9).

- A few of the gene ontology graphs have lines overlapping that make them challenging to read

We thank the reviewer for spotting this – we have now improved the display of the gene ontology graphs in the revised manuscript.

- Supp Fig 5j seems to have an incorrect description in the figure legend that doesn't match with what appears in the figure

We thank the reviewer for having spotted this, we had inadvertently kept this description from an initial draft of the figures. This is now corrected.

- Fig 4E: Given these results, it seems relevant for the authors to characterize the percent of accessible regions that are at the TSS, intergenic, and intragenic for each of the 7 categories in this figure.

We thank the reviewer for this suggestion. We have now performed the requested analysis, which is available as Figure EV5J. We compared the feature overlap (intergenic, transcription termination

site, transcription start site/promoter, exon, and intron) annotated in the mouse genome mm10/GRCm38 in the seven groups and found that as expected, the downregulated loci were enriched at promoters.

Reviewer #2:

Strebinger et al. study how variations in the levels of the pluripotency factors Oct4 and Sox2 impact the differentiation of mouse embryonic stem cells to neuroectoderm or mesendoderm. The authors have an interesting set of ES lines in which they can monitor Oct4, Sox2 and two differentiation markers (Sox1 and Brachyury) in single cells or vary pluripotency factor levels in an inducible manner. They find that Sox2 represses its own expression and induces Oct4 expression. As Sox2 and Oct4 form heterodimers, it would be of particular interest to study the correlation between Sox2 and Oct4 fluctuations in single cells and the impact of differential Sox2/Oct4 levels on differentiation (both of which the authors can easily do with their cell line). The authors also suggest that Sox2 and Oct4 levels influence the commitment of ES cells. This part of the study is potentially very interesting but is supported by much weaker and contradicting evidence.

We thank the reviewer for her/his interest in our work. Importantly, we would like to specify that while SOX2 indeed represses the transcription of its own gene, SOX2 does not “induce Oct4 expression” (Fig 2B), but increases the stability of the OCT4 protein (Fig 2C). Thus, the reviewer might be misled in thinking that an increase in SOX2 protein levels will modulate Oct4 transcription. Oct4 transcription was shown to occur in a pulsatile, bursting manner (Skinner et al., eLife 2016), and is therefore likely to be an important if not the main contributing factor to OCT4 protein fluctuations. Since high SOX2 protein levels increase OCT4 protein half-life but not Oct4 transcription, this will lead to an increase in overall OCT4 levels but there is no reason to expect clear changes in the temporal pattern of OCT4 fluctuations as a function of SOX2 levels, which in addition would be extremely hard to determine since both proteins fluctuate on similar timescales. Importantly, in Fig 2E we show that cell populations sorted for high/low endogenous SOX2 levels increase/decrease their OCT4 levels, respectively, thus showing that SOX2 fluctuations do impact OCT4 levels – i.e. cells with higher SOX2 levels have a higher probability to increase than to decrease their OCT4 levels within 8 hours. In contrast, we found that OCT4 fluctuations do not impact SOX2 levels (Fig 2F), which is also expected since OCT4 overexpression did not alter SOX2 protein levels (Fig EV2C).

The reviewer suggests to study « the impact of differential Sox2/Oct4 levels on differentiation ». In our study, we have actually done this by sorting cells for different levels of SOX2, OCT4 and SOX2/OCT4 ratios, now using three different differentiation systems and totalling 19 independent biological replicates (Fig 3H-L). Note that sorting cells into SHOH, SHOL, SLOH and SLOL populations allows addressing the impact of both overall SOX2 and OCT4 levels and of the SOX2/OCT4 ratio (even though we can only measure relative SOX2/OCT4 ratios since we cannot determine absolute copy numbers of in flow cytometry measurements of SOX2-SNAP and OCT4-HALO). Importantly, both SHOH and SLOL cell populations have similar SOX2/OCT4 ratios, but they have very distinct differentiation potentials in undirected differentiation conditions (Fig 3I) and to a lesser extent in directed differentiation towards the mesendoderm (Fig 3K). This suggests that the absolute levels of SOX2 and OCT4, rather than the SOX2/OCT4 ratio impacts differentiation towards the neuroectoderm and mesendoderm. We have now clarified this on page 13 of the revised manuscript.

We disagree that our study on how SOX2 and OCT4 influence the commitment of ES cells is supported by « much weaker and contradicting evidence ».

Concerning the « weaker » evidence, in the context of undirected differentiation we consistently show that:

1. Elevated endogenous SOX2 levels at the onset of differentiation increase neuroectodermal commitment. These conclusions are driven from both time-lapse imaging

of individual cells (Fig 3C) and from 10 independent experiments of FACS/flow cytometry experiments (Fig 3E and 3I). Our conclusions are backed up by strong statistical evidence.

2. Elevated endogenous OCT4 levels at the onset of differentiation strongly increase both neuroectodermal and mesendodermal commitment from 9 independent experiments of FACS/flow cytometry experiments (Fig 3F and 3I). Here again, our conclusions are backed up by strong statistical evidence.

Concerning the « contradictory » evidence, since the reviewer did not point at what she/he judges as contradictory, we can only guess that this refers to the impact of OCT4 levels on neuroectodermal differentiation. We have found that endogenously high OCT4 levels before differentiation result in increased neuroectodermal commitment. This contrasts with OCT4 overexpression or high endogenous OCT4 expression levels throughout differentiation, which results in inhibition of neuroectodermal commitment (Thomson et al., Cell 2011). Importantly, our own experiments of OCT4 overexpression throughout differentiation are in complete agreement with previous reports (Fig EV4E or our manuscript showing similar findings than Thomson et al., Cell 2011). Thus, high OCT4 levels have opposite effects on neuroectodermal commitment depending on the context in which OCT4 levels are high (before vs during differentiation) and therefore we conclude that the impact of OCT4 levels on neuroectodermal differentiation is highly context-dependent. We have now clarified this in the discussion section on page 17 of the revised manuscript.

Major concerns:

1. The differentiation experiments show extensive variability in the fraction of Brachyury positive cells between assays. These differences cannot be explained by the sorting schemes themselves. Indeed in Supplementary Fig. 4a, 62 to 78% of cells are Brachyury positive whereas control of other assays have fewer Bra+ cells. It is also worrisome for the reproducibility of the assays that the no Dox controls in Supplementary Fig. 4c and 4d are very different. In order to believe the conclusions, the data should show internal consistency between controls.

Here it is important to keep in mind that differentiation heterogeneity of mouse embryonic stem cells is a well-known general issue (Torres-Padilla and Chambers, Development 2014). In addition, we on purpose used an undirected differentiation strategy to assess the impact of OCT4 and SOX2 fluctuations on ES cell differentiation in an unbiased manner, and this strategy results in a particularly large variability in the amount of mesendodermal or neuroectodermal cells obtained. This suggests that there are many unknown parameters varying from one experiment to another that can bias differentiation outcomes. However this does not compromise the robustness of our findings, which are backed up by strong statistical significance (Fig 3E, F and I). Our conclusions on the impact of OCT4 and SOX2 on mesendodermal and neuroectodermal commitment in the context of undirected differentiation shown in Fig 3E, F and I were made from at least 9 independent experiments for each population (at least 4 for data shown Fig 3E and F, and 5 for Fig 3I). This amounts in total to (at least) 14 comparisons between cell populations with either high or low OCT4 or SOX2 levels. Also note that the impact of SOX2 on neuroectodermal differentiation is further substantiated by our quantitative time-lapse imaging data (Fig 3C).

2. Halo-TMR emission spectrum overlaps significantly with mCherry's, questioning its use in the SBROS line. The authors should demonstrate that the labeling schemes they use for the SNAP and Halo tags don't interfere with the measurement of mCherry by using other substrates (for example a blue one).

Let us clarify this. Using a blue dye would prevent us from sorting cells in G1 or S phase using Hoechst staining, and Halo dyes with other excitation/emission would overlap with either eGFP or SiR647. However, as we explain below, we can easily demonstrate that Halo-TMR labelling of OCT4 does not interfere with mCherry measurements after differentiation. Also note that an interference between TMR and mCherry signals after several days of differentiation is highly unexpected, given the half-life of Halo-OCT4 and the numerous cell divisions that occur and further dilute the dye before flow cytometry analysis.

We have now performed additional experiments of sorting after staining with both Halo-TMR dye and SNAP-SiR 647 dyes, in which we directed differentiation towards neuroectoderm. In this context we have only very little ME differentiation, thus any residual TMR dye should be detected in the mCherry channel and be stronger in cells initially sorted for high levels of OCT4 using the TMR dye. On average, we found 0.50% of mCherry-positive cells upon directed NE differentiation of OCT4-high cells, and 0.58% of mCherry-positive cells upon directed NE differentiation of OCT4-low cells (values calculated from Appendix Table S10). Therefore, this excludes any major bias in our mCherry readout due to the presence of residual TMR dye.

3. The authors find that high Oct4 levels enhance neuroectoderm commitment in contradiction with a previous report. To further substantiate their claim, they should study how Oct4 and Sox2 fluctuations impact directed differentiation to neuroectoderm. It will also add generality to the authors' findings.

We guess that the « contradiction » the reviewer mentions refers to previous work claiming that OCT4 decreases neuroectodermal commitment when endogenously high or constantly overexpressed during differentiation (Thomson et al., Cell 2011). Importantly, in our differentiation system we also observed that OCT4 overexpression during differentiation decreases neuroectodermal commitment (Fig EV4E). However, our work focused mainly on the impact of endogenous OCT4 levels before the onset of differentiation, which results in increased neuroectodermal commitment. Importantly, our conclusions are made from comparing neuroectodermal commitment for OCT4-high vs OCT4-low cells (Fig 3F), OCT4-high/SOX2-high cells vs OCT4-low/SOX2-high cells (Fig.3I) and OCT4-high/SOX2-low cells vs OCT4-low/SOX2-low cells (Fig.3I), thus corresponding in total to 14 different comparisons, and each comparison group was backed-up by strong statistical significance. Therefore, we do not think that the fact that OCT4 levels before the onset of differentiation do or do not impact neuroectodermal commitment in a directed differentiation system will have direct implications on the validity of our claims. Importantly, differentiation signals may override the impact of OCT4 on neuroectodermal differentiation, similarly to what we observed for the impact of OCT4 levels on mesendodermal commitment upon directed mesendodermal differentiation (Fig.3K).

Nevertheless, we agree that studying the impact of OCT4 levels in the context of directed neuroectodermal differentiation should deepen our understanding of the context-dependency of the effect of OCT4 levels on cell fate decisions. We have now performed the suggested experiments in 7 biological replicates. We sorted ES cells in the G1 phase of the cell cycle and in SHOH, SHOL, SLOH, and SLOL populations, followed by differentiation for 4 days in the presence of 1 μ M of SB-431542 (a TGF- β receptor inhibitor) and 25 ng/ml bFGF, which almost completely abolished mesendodermal commitment (0.54% of mCherry-positive cells on average after 4 days of differentiation, Fig EV4J) and increased the efficiency of neuroectodermal commitment (Fig EV4J). In this context, high OCT4 levels did not significantly increase or decrease neuroectodermal commitment (Fig 3L). Therefore, this confirms that high OCT4 levels before induction of differentiation, even if directed towards the neuroectodermal lineage, do not inhibit neuroectodermal commitment. This contrasts with OCT4 overexpression or high endogenous OCT4 levels throughout differentiation, which inhibit neuroectodermal commitment (Thomson et al., Cell 2011). It also shows that while high OCT4 levels at the time of 2i and LIF removal enhance neuroectodermal commitment in the context of undirected differentiation (Fig 3F and I), this impact can be overridden by differentiation signals. This is similar to the absence of impact of high OCT4 levels on mesendodermal commitment upon directed mesendodermal differentiation (Fig 3K). We have now further discussed this on page 17 of the revised manuscript.

4. While Sox2 has a constant production rate during the cell cycle, Oct4 has an interesting production profile (Supplementary Fig. 3). The data in Fig. 2j suggests quite some heterogeneity between single cells.

We thank the reviewer for this comment. We would like to clarify that we did neither measure nor mention the « production profile » of OCT4 or SOX2 in our manuscript. The temporal changes in total OCT4 or SOX2 levels (or of their concentrations) depend on both their synthesis rates and

degradation rates. While we observed a modest increase in OCT4 concentrations in early G1, these changes were not significant at the population level.

What is the heterogeneity in the production rate of Sox2 and Oct4 during the cell cycle in single cells?

Here we would like to clarify that we did not and cannot determine SOX2 and OCT4 production rates. It would require measuring protein synthesis and degradation rates simultaneously using a completely different set of tools, such as for example a fluorescent timer strategy as we did in Alber et al., *Molecular Cell* 2018, which would require the generation and validation of new knock-in cell lines and is thus out of the scope of this study.

However, we can comment on the variability in the concentrations/total levels of SOX2 and OCT4 in individual cells, which is illustrated by the shaded areas in Fig EV3A, B, E and F. There we can appreciate that there are no substantial differences in concentration variability of either SOX2 or OCT4 across the cell cycle.

How do the production rates of Sox2 and Oct4 correlate in single cells?

Again we cannot measure the production rates of OCT4 or SOX2, so we cannot comment on this.

How does the ratio between Sox2 and Oct4 vary in individual cells during the cell cycle?

This is an interesting point. To satisfy the reviewer's request, we have now determined the ratios of SOX2 and OCT4 in different cell cycle phases using flow cytometry. Analyzing SBROS cells stained with Halo-TMR and SNAP 647-SiR in G1, S, and G2/M phase cells based on Hoechst reveals a cloud of values representing different combinations of SOX2/OCT4 levels. While the absolute SOX2 and OCT4 levels increase during cell cycle progression due to the growth of the cells, the SOX2/OCT4 ratios are very similar across the cell cycle (Figure 2 below). The Pearson's correlation (R) coefficients are also similar across the cell cycle (0.32 in G1, 0.37 in S, and 0.42 in G2M), and their low value reflects the noisy positive correlation between SOX2 and OCT4 levels.

Figure 2: Correlation between SOX2-SNAP and OCT4-HALO levels at different cell cycle stages.

5. Does the Oct4/Sox2 ratio influence commitment?

This is an important question, that we in fact addressed by measuring the differentiation potential of SHOH, SHOL, SLOH and SLOL cells. The SOX2/OCT4 ratio is similar in SHOH and SLOL cells, but these cells have a very distinct differentiation potential. We found that the level of OCT4 (in the context of undirected differentiation) is the dominant determinant of differentiation outcome, with a more modest impact of the level of SOX2 (Fig 3I). Thus, the SOX2/OCT4 ratio does not have a prominent role in directing the NE and ME differentiation decisions. We have now discussed this on page 13 of the revised manuscript.

Minor concern:

The readability of Fig. 2 and 3 could be improved by choosing more contrasting colors and adding extra labels.

We thank the reviewer for her/his comments. We have now improved the display and labelling of Fig 2 and 3 in the revised manuscript.

2nd Editorial Decision

30th August 2019

Thank you for sending us your revised manuscript. We have now heard back from the two reviewers who were asked to evaluate your revised study. As you will see below, the reviewers are satisfied with the modifications made and think that the study is now suitable for publication in Molecular Systems Biology.

Before we formally accept your manuscript for publication we would ask you to address a few remaining editorial issues listed below.

REFeree REPORTS

Reviewer #1:

The authors address successfully most of my comments with new experiments and analyses as well as with providing a better and sufficient clarification of the previously confusing points. The revised manuscript has been significantly improved in terms of clarity and interpretation of results. Although, few comments remain not sufficiently addressed, I believe that the overall quality and relevance of this work justifies publication in Molecular Systems Biology.

Reviewer #2:

The authors have satisfactorily answered my concerns in their revision.

Corresponding Author Name: David Suter

Manuscript Number: MSB-19-9002